# Rethinking Craig and Gordon's approach to modeling isotopic compositions of marine boundary layer vapor

Xiahong Feng[1], Eric S. Posmentier[1], Leslie J. Sonder[1], Naixin Fan[1.2]

[1]Department of Earth Sciences, Dartmouth College, Hanover, NH 03755, USA
[2]Current address: Department Biogeochemical Integration, Max Planck Institute for Biogeochemistry, Hans-Knöll-Straße 10 07745, Jena, Germany

*Correspondence to*: Xiahong Feng (Xiahong.feng@dartmouth.edu)

**Abstract.** We develop a one-dimensional (1D) steady state isotope marine boundary layer (MBL) model that includes meteorologically important features missing in Craig and Gordon type models,
namely height-dependent diffusion/mixing, lifting to deliver air to the free troposphere, and convergence of subsiding air. Kinetic isotopic fractionation results from this height-dependent diffusion that starts as pure molecular diffusion at the air-water interface and increases with height due to turbulent eddies. Convergence causes mixing of dry, isotopically depleted air with ambient air. Model results fill a quadrilateral in $\delta D$-$\delta^{18}O$ space, of which three boundaries are respectively defined by 1)
vapor in equilibrium with various sea surface temperatures (SSTs); 2) mixing of vapor in equilibrium with seawater and vapor in subsiding air; and 3) vapor that has experienced maximum possible kinetic fractionation. Model processes also cause variations in d-excess of MBL vapor. In particular, mixing of relatively high d-excess descending/converging air into the MBL increases d-excess, even without kinetic isotope fractionation. The model is tested by comparison with seven datasets of marine vapor
isotopic ratios, with excellent correspondence. About 95% of observational data fall within the quadrilateral predicted by the model. The distribution of observations also highlights the significant influence of vapor from nearby converging descending air on isotopic variations within the MBL. At least three factors may explain the ~5% of observations that fall slightly outside of the predicted regions in $\delta D$-$\delta^{18}O$ and d-excess-$\delta^{18}O$ space: 1) variations in seawater isotopic ratios, 2) variations in isotopic
composition of subsiding air, and 3) influence of sea spray.

# 1 Introduction

Stable isotopic ratios of water have been widely used to study the hydrologic cycle of the atmosphere. They have proven to be a powerful tool for understanding modern atmospheric processes (e.g., Dansgaard, 1964, Lawrence et al., 2004; Worden et al., 2007; Uemura et al., 2008; Kurita, 2011; Kopec et al., 2017). In addition, they have been extremely useful for inferring paleoclimate conditions and making climate reconstructions from glacier ice, tree rings, lake sediments, speleothems, and paleosols (e.g., Dansgaard et al., 1989; Wang et al., 2001; Huang et al., 2002; Andersen et al., 2004; Jouzel et al., 2007; Feng et al., 2007; Sheldon and Tabor, 2009; Masson-Delmotte et al., 2015).

Sound interpretation of isotopic data requires a thorough understanding of all processes in the hydrological cycle that affect isotopic variations. These include 1) surface evaporation and processes in the planetary boundary layer (PBL) through which vapor reaches the overlying free atmosphere; 2) rainout and other processes along the trajectory of air masses transported to a precipitation site; 3) nucleation, growth, coalescence, and reevaporation of hydrometeors between the moisture source area and the precipitation site; and 4) subsequent processes affecting precipitation as it falls through the air. This study focuses on the first of these – surface evaporation and isotopologue concentrations within and fluxes through the PBL --- in particular, the marine boundary layer (MBL), where ascending air delivers water vapor to the free atmosphere.

The PBL and the MBL have a variety of qualitative and quantitative definitions, not all consistent. In this discussion, we use the phrase "Boundary Layer" to refer to the lower part of the planetary or marine atmosphere, in which the flux of water vapor is close to vertical and vapor transport is accomplished primarily by turbulent or convective mixing. The troposphere above the MBL is often referred to as the "free atmosphere" or "free troposphere", in which vapor transport is dominated by near-horizontal advection by winds. The thickness of the MBL varies from ~100 m to ~1000 m or more with location, season, and time of day, as well as weather conditions (e.g., Christakos et al., 2013; Winning et al., 2017). In the MBL, unlike the terrestrial part of the PBL, water vapor is not affected by plant transpiration or variable surface wetness.

Craig and Gordon (1965) developed the first isotopic evaporation model, referred to hereafter as the C-G model, to calculate isotopic ratios of the evaporation vapor flux from the water surface when the humidity and isotopic composition in the "free air" are specified. The C-G model is based on the diffusive flux of an isotopologue (e.g., $H_2^{16}O$, or HDO) through the boundary layer of the atmosphere.
The diffusive flux is proportional to the difference in isotopic composition of vapor at the layer's boundaries and inversely proportional to the resistance of the layer to transport (Fick's Law, described as a simple analogy of Ohm's Law by C-G). The C-G model is conceptually a multiple "slab" (0-dimensional) model. The slabs (layers), stacked from the bottom up, are turbulent ocean water, a laminar layer of ocean water that is affected by evaporation, the water-air interface, a laminar layer of
air, a turbulent air layer, and the free atmosphere (where humidity and isotopic ratios no longer change rapidly with height). Even though each layer has a different resistance to vapor transport, the fundamental premise is that the vapor flux through all layers is the same. This premise follows in turn from the assumptions of quasi steady state conditions, conservation of mass, and zero horizontal fluxes. Therefore, the flux entering the free atmosphere (at the top of the PBL) equals the evaporative flux at
the water surface. The C-G model was tested and empirically parameterized using measurements of the isotopic evolution of an isolated body of evaporating water (Craig et al., 1963), and was subsequently successfully applied and adapted to many specific applications, including lake evaporation, leaf transpiration, and marine boundary layer processes. Interested readers can refer to Horita (2008) for a comprehensive review of the status of the C-G model.

Particularly relevant to this study is the adaptation of the C-G model for the marine boundary layer. An influential study by Merlivat and Jouzel (1979) linked the magnitude of kinetic isotopic fractionation primarily within the laminar layer above the water-air interface, which is required input for the C-G model, to aerodynamic conditions, i.e., wind speed and surface ocean roughness. However, the model still required the input of the free atmosphere humidity and isotopic ratios. Recognizing the difficulty of
knowing these free atmosphere variables, Merlivat and Jouzel (1979) made an assumption, known later as the "closure assumption", that the isotopic ratios of *vapor mass* in the free atmosphere were equal to the isotopic ratios of the *vapor fluxes* from the sea surface. This assumption enabled them to complete a new multi-slab model (the M-J model), used by numerous investigators to calculate isotopic fluxes from

the sea surface over a range of maritime conditions, and to explore relationships between isotopic compositions of evaporative flux and boundary layer meteorological conditions such as sea surface temperature and relative humidity (e.g., Johnsen et al., 1989; Petit et al. 1991). The closure assumption also allowed the modeled flux to be used as the starting isotopic composition of an air mass, which evolves during subsequent transport and rainout or a Rayleigh process (e.g., Johnsen et al., 1989; Petit et al. 1991). The closure assumption, however, has been determined to be generally invalid at local scales (Jouzel and Koster, 1996). Nevertheless, it has continued to be used (e.g., Benetti et al., 2014) simply for lack of a better assumption.

Abandoning this closure assumption requires a fundamental rethinking of the MBL model structure. In addition, there are ramifications of other model assumptions. As a consequence, we consider three requirements for developing a useful, physically consistent MBL model free of the invalid closure assumption, a model with the purpose of determining the isotopic ratios of air ascending from the top of the MBL and entering the free atmosphere above.

First, vertical advection is necessary at the inception of a Rayleigh process in order to lift MBL air into the free atmosphere. When an air mass is lifted into the free troposphere, the vapor isotopic ratio of the air that first condenses during the Rayleigh process is equal to the isotopic ratio of vapor within the air mass, not the ratio of isotopologue diffusive *fluxes* into the air mass. Contrary to the closure assumption, these two are not generally equal (Jouzel and Koster, 1996). Therefore, an MBL model should calculate not only the isotopic ratio of vapor flux at the sea surface, but also that of vapor concentration within the MBL, particularly at the top of the MBL, and the latter is the quantity that should be used for the initial vapor isotopic composition in any subsequent Rayleigh process.

Second, with incorporation of vertical advection, mass balance requires 1) horizontal convergence of air within the MBL to replenish the lifted air in the evaporation column, and 2) subsidence of air outside the model region to sustain the local horizontal convergence. Such a circulation on various scales was discussed by Craig and Gordon (1965) to explain why vapor in the MBL was not in isotopic equilibrium with ocean water. In this contribution, we attempt to quantify how horizontal convergence of air from non-local regions of subsidence affects the isotopic properties of the local MBL. Because the

converging air is unlikely to have the same isotopic composition as the local MBL air, convergence turns out to affect MBL vapor isotopes quite significantly, as discussed later in this paper.

Third, incorporating convergence means that the assumption of constant vertical flux in the C-G model is no longer valid, even under steady state conditions. This necessitates different equations of mass conservation.

These three model requirements obviously require rethinking the C-G approach to constructing a model, and draw a sharp divergence between our model and the models of C-G and its extensions (e.g., Jouzel and Merlivat, 1979; Benetti et al, 2015). In addressing these three required changes, we find it advantageous to incorporate two additional changes to the model structure. The fourth major change is to abandon previous multi-slab models (referred to hereafter as C-G type models) and to adopt instead a true one-dimensional (1D) model in which quantities such as flux change continuously with height. In doing so, we describe a coefficient of turbulent transport, that increases with height (see below). This yields the additional benefit of the ability to obtain isotopic ratios of air and vapor flux at any given height within the MBL.

The fifth major change is to abandon the need to specify the kinetic fractionation factor required by C-G type models. In the boundary layer, the eddy diffusion coefficient (coefficient of turbulent transport by eddies) increases continuously with height (Merlivat and Coantic, 1975) from zero at the air-water interface, where transport of vapor is affected solely by isotopically fractionating molecular diffusion, to greater values at heights where vapor transport is dominated by turbulent eddies. Such a height-dependent change of diffusion coefficient is adopted in our model. As a result, our model is relieved of the need 1) to empirically choose the value of a parameterized kinetic fractionation factor ($\Delta\varepsilon$ in Craig and Gordon, 1965; $k$ in Merlivat and Jouzel, 1979; this choice may sometimes be difficult and values reported in the literature may not apply to the specific conditions under investigation (e.g., Xiao et al., 2017)), and 2) to specify a specific laminar layer thickness. Instead, we allow the total diffusion coefficient to represent pure molecular diffusion at the interface (which differs for different isotopologues and thus leads to kinetic fractionation), and to increase linearly with height to several orders of magnitude greater than molecular diffusivities. As a result, the laminar layer thickness scale

becomes a diagnostic variable. The fourth and fifth major changes further sharpen the distinction between our model and C-G type models.

Abandoning some of the assumptions of earlier models, such as constant flux of vapor isotopologues, flux equal to concentration (the invalid closure assumption), and presence of a discrete laminar layer, permits a significantly more realistic and elucidative approach to understanding processes in the MBL and allows more meteorological profiles of variables (such as humidity and isotopic ratios of the vapor in the MBL) to be calculated rather than specified. The trade-off, obviously, is in sacrificing the simplicity of the classical model. The model reported here attempts to balance that trade-off: it is considerably less complex than isotope-enabled general circulation models (GCMs), or other three dimensional, all-inclusive boundary layer models (e.g., Wei et al., 2018) and should be accessible to investigators without substantial experience with complex models, yet it allows exploration of physical controls of vapor within the MBL and in the initial Rayleigh process above the MBL.

The model introduced here is a one-dimensional (vertical) steady state model with three stratified layers within the MBL. It adopts the following enhancements to improve upon the earlier, classical models: 1) It explicitly includes vertical velocity and horizontal convergence of air and vapor, notwithstanding the difficulties of specifying the fluxes and isotopic properties of converging air. 2) It uses a height-dependent eddy diffusion coefficient without increasing the total number of free parameters (degrees of freedom) in the model. 3) It does not make the closure assumption that isotopic flux equals isotopic composition. 4) It solves not only for isotopologue fluxes, but also concentrations. 5) MBL humidity and kinetic fractionation factors are no longer required input parameters but are calculated. 6) Vapor fluxes are no longer constant with height.

Above, we have made several references to applying an MBL model to the initiation of a Rayleigh model of vapor trajectories in the free troposphere, but there is another crucial role for an MBL model. It is the model's application to understanding the systematics linking isotopic observations of precipitation to the meteorological conditions of the vapor source, of the precipitation site, and along the moisture paths between the two. We use the new MBL model presented here to examine the vapor source part of the isotope systematics. Since the model produces vapor concentrations and isotopic

ratios, it can be tested and validated by MBL isotopic measurements, which, thanks to new spectral vapor isotopic measurement technology, have become increasingly available. There are still additional potential benefits. For example, such a model might provide a new way to estimate evaporation rate, one of the holy grails of weather and climate models.

In the following sections, we first describe the formulation and solution of the model and the marine boundary layer observations to be used to validate the model. Then we discuss the model results and their comparison with the observations, as a basis for addressing the systematics of vapor source conditions and atmospheric isotopes. Although the limitations of the model will be discussed in more detail in section 6.2, we briefly mention here that this model applies to the part of the marine boundary

layer where vertical velocity is positive (upward), there is no net horizontal advection, and the model does not include vapor liquid exchange within the air column.

**2 The Isotope Marine Boundary Layer Model**

The model we describe here has been developed to study the effect of marine boundary layer processes, such as evaporation of water, mixing and uplift of air, on concentrations and fluxes of isotopologues of

the MBL. Three isotopologues, $H_2^{16}O$, $H_2^{18}O$ and HDO are modeled and presented here, but more can be added easily. We refer to this model as the Isotope Marine Boundary Layer (IMBL) model.

Figure 1 is a cartoon of the IMBL model showing the three layers that comprise the model column itself, and the input of external air. Layer 1, the lowest layer, extending from the surface at $z=0$ to height $z=h_1$, is a quasi-von Kármán layer in which vapor is transported upward from the sea surface by mixing

that increases in intensity with height. Layer 2 ($h_1<z<h_2$), the middle layer, is subject to strong vertical mixing, to the convergence of air that has elsewhere descended from the free atmosphere and converged horizontally into the modeled column, and to vertical advection caused by the convergence.  In Layer 3 ($h_2<z<h_3$), the top layer, there is no convergence, so the air ascends at a fixed rate, while the vertical mixing rate decreases in intensity with height.

Sketched on the right side of Figure 1 are vertical profiles of the diffusion coefficient *K(z)* and the (dynamic) vertical velocity *w(z)*. The profile of *K(z)* is consistent with typical variation with height of

the eddy viscosity diffusion coefficient in the boundary layer, based on O'Brien (1970). The coefficient $K(z)$ equals the molecular diffusion coefficient $K_m$ at the surface and increases linearly with height to a maximum value $K_{max}$ at $z=h_1$. It remains fixed at $K_{max}$ through the middle layer, then decreases linearly in the top layer above $z=h_2$ to a small value $K_t$ at $z=h_3$. The vertical velocity $w(z)$ is zero in Layer 1, increases linearly with height through the middle layer, in which convergence occurs at a fixed rate, and remains constant at value $w_a$ in Layer 3. Consistent with their constant values of $w$, Layers 1 and 3 do not have convergence.

The following subsections, 2.1-2.3, describe the individual physical and mathematical features of the model. Table 1 contains a list of variables and parameters found in these subsections and elsewhere.

## 2.1 Mixing Process

The central matter for this subsection is the specification of the height-dependent eddy diffusion coefficient, $K_i(z)$, which appears in Fick's Law for diffusive flux,

$$F_i = -K_i(z)\frac{\partial(\rho C_i)}{\partial z}, \qquad (1)$$

where $F_i$ is the vertical flux of the $i$'th isotopologue (isotopologue-mass area$^{-1}$ time$^{-1}$), $C_i$ is the concentration ratio of the $i$'th isotopologue (isotopologue-mass dry-air-mass$^{-1}$), $\rho$ is the density of dry air (mass-of-dry-air volume$^{-1}$), and $z$ is the vertical coordinate (increasing upwards from $z=0$ at the surface). The $i$-subscripts of $F$, $K$ and $C$ are reminders that they all depend on the specific isotopologue under consideration, but for simplicity we drop them hereafter. We note that $C_i$ has the same units as the commonly used mixing ratio $r$. The difference is that $r$ is total water vapor mass per unit dry air mass, while $C_i$ is the mass of the $i$'th isotopologue (e.g., $H_2^{18}O$) per unit dry air mass. In this paper, we will use the term concentration ratio for $C$ and mixing ratio for $r$. With (1), we assume that Fick's Law can be used to represent vertical mixing by the combined effects of mechanically-driven turbulence, buoyancy-driven convection, and molecular diffusion.

In adopting Fick's Law, here, we have made the tacit assumption that alternative mixing models are less appropriate for our purposes. While higher-order closure schemes (e.g., Burk, 1977), structured

turbulence models (e.g., Kirwan, 1968), and the telegraph equation (e.g., Goldstein, 1951) have some advantages over Fick's Law, their added complexity would not be justified at this juncture, and we postpone their consideration until future investigations warrant.

Conservation of mass for an isotopologue affected only by diffusion, temporarily neglecting convergence and advection, takes the form

$$\frac{\partial(\rho c)}{\partial t} = -\frac{\partial F}{\partial z}. \tag{2}$$

For $F$ given by Eq. (1), and for the case of $\rho$ with negligible dependence on $z$ or $t$, Eq. (2) becomes

$$\frac{\partial c}{\partial t} = \frac{\partial}{\partial z}\left(K(z)\frac{\partial c}{\partial z}\right). \tag{3}$$

Returning to the central matter, the specification of $K(z)$, we reject the assumption of constant $K$, the simplest and most frequently used assumption, because it is particularly unrealistic near boundaries (i.e., water-air interface in this work), where the inhibitive effect of the interface on mixing of air increases with proximity to the boundary. The next most frequently used assumption is that $K$ is a linear function of $z$, although a few others have been proposed (Merlivat and Coantic, 1975). The use of linear functions of $z$ to represent $K(z)$ has a long history in turbulence studies, including the turbulent transport of momentum as well as both buoyantly active and passive scalar fluid properties. The well-known work of von Kármán (1930) and Prandtl (1932) successfully applied the simple form $K(z) = b \cdot z$, where $b$ is a constant, to derive the equation of the logarithmic layer, where $u(z) = s \cdot \ln(z) + m$, with $u$ being the wind speed, and $s$ and $m$ being constants.

An obvious limitation of the widely cited von Kármán/Prandtl formulation occurs when $z$ is very small, near the singularity at $z$=0. The most common way to circumvent this problem has been to introduce a discrete "laminar boundary layer" (LBL), a very thin but finite layer with constant diffusion by molecular motion and with weak turbulent influence. The incremental cost of this approach is the necessity of specifying one additional parameter, δ, the thickness of the LBL.

Another way to overcome the problem for small $z$ is to use the more general form:

$$K(z) = K_m + b \cdot z, \tag{4}$$

where $K_m$ is the molecular diffusion coefficient for vapor in air and $b \cdot z$ is the contribution of turbulent eddies to the diffusion coefficient. An equivalent general linear form was applied to boundary layer mixing above the LBL by Montgomery (1940), and within the LBL by Sverdrup (1946; 1951). Note that $K_m$ varies among isotopologues, but $b$ does not. This is the basic cause of kinetic fractionation. When $z$ is small (see Eq. (4) and $z<z^*$, below), the relative differences among $K(z)$ values for different isotopologues are large, which is the basis for strong kinetic isotopic fractionation near the interface.

One advantage of the form of Eq. (4) for the parameterization is the gain of one degree of freedom through the use of the known quantity $K_m$ instead of the unknown parameter $\delta$, the thickness of the laminar layer. The latter can be replaced by a diagnostic laminar layer thickness, $z^*$, the height where molecular and turbulent diffusion coefficients are equal. In other words, below $z^*$ vertical diffusion is dominated by molecular processes, and above it turbulence and convection prevail. From Eq. (4),

$$b \cdot z^* = K_m. \tag{5}$$

The $z^*$ values reported in this paper were computed using the diffusion coefficient of $H_2^{16}O$. A linear approach, mathematically equivalent to Eq. (4) (Sheppard, 1958), used bulk aerodynamic theory to modify Sverdrup's (1946, 1951) work. The result was another linear function of $z$ containing the friction velocity $u^*$ and von Kármán's constant $\kappa$ instead of the coefficients $K_m$ $and$ $b$, thus connecting Sheppard's model to familiar parameters of fluid mechanics.

Merlivat and Coantic (1975) tested and compared various linear and nonlinear alternatives to Eq. (4). In contrasting alternative boundary layer models for use in isotope studies, they concluded that their laboratory experiments did not support Sheppard's linear theory. However, at the larger scale of Arctic lake field experiments, Eq. (4), which is mathematically equivalent to Sheppard's (1958) approach, was used successfully to model atmospheric vapor isotopes (Feng et al, 2016).

There are several additional benefits to using Eq. (4) rather than parameterizing the kinetic isotopic fractionation. First, $K$ is a boundary layer dynamics parameter that already exists in boundary layer dynamics literature. More importantly, by making kinetic fractionation a function of $K(z)$, our model is capable of exploring how boundary layer mixing affects the kinetic isotopic fractionation. In addition, this formulation allows our model to compute fluxes of isotopologues, not just their ratios, which in turn allows computation of sea surface evaporation. This is significant because the isotopic distribution can then be used to constrain evaporation rate (Feng et al., 2016). Furthermore, with $K$ a continuous function of $z$, our model is truly one-dimensional, which allows vertical isotopic profiles to be predicted and compared with isotopic observations at multiple heights and with any resolution (Feng et al., 2016). Hence, we proceed with Eq. (4).

## 2.2 Convergence and Vertical Advection

Moist air undergoing the Rayleigh distillation process in the free atmosphere is generally conceived to have originated in the PBL and been lifted (i.e., vertically advected) into the free atmosphere. For mass to be conserved, such uplift must be accompanied by convergence within the PBL. Nevertheless, C-G type models ignore convergence within the boundary layer (e.g., Craig and Gordon, 1965; Merlivat and Jouzel, 1979). The incorporation of this apparent contradiction into a model might be justified by arguing that the effect of boundary layer convergence on isotopic processes is negligible, or if the only concern is the isotopic evolution of the liquid where the vapor originates. In the IMBL model presented here, however, we choose to preserve consistency by including both convergence and uplift, and to use model results to diagnose the importance of the convergence effect rather than neglecting it *a priori*. As we later show, convergence has a large influence on the isotopic composition of the air exiting the MBL upward into the free atmosphere.

Steady-state conservation of mass for dry air, using dynamic variables and neglecting diffusion, can be written in the form

$$D - \frac{\partial}{\partial z}\big(w(z)\big) = 0 \qquad (6)$$

where $D$ (here considered independent of $z$) is the horizontal dynamic convergence (dry-air-mass volume$^{-1}$ time$^{-1}$), and $w(z)$ is the dynamic vertical velocity (dry-air-mass area$^{-1}$ time$^{-1}$), which is positive for upward air movement. The kinematic (conventional) velocity (length time$^{-1}$) is the dynamic velocity divided by the air density, $\rho$. Eq. (6) indicates when $D$ is positive, $w$ increases upward. We will use Eq. (6) to derive the governing equation for the middle layer in section 2.3.2.

Ignoring (for now) the effect of diffusion, conservation of mass for isotopologues affected only by kinematics can now be expressed as

$$\rho \frac{\partial C}{\partial t} = D(C_C - C) - w(z)\frac{\partial C}{\partial z},$$ 
(7)

where $C_C$ is the concentration ratio of the isotopologue of the MBL air converging into the area being modeled. The first and second terms on the right are the direct effect of convergence (replacement of air of concentration ratio $C$ by converging air of concentration ratio $C_C$) and the effect of vertical advection, respectively. Note that Eq. (7) is also consistent with the assumed absence of non-divergent horizontal advection.

The converging air, with concentration ratio $C_C$, is a mixture of two air types, with fractional presence by mass $\beta$ and (1-$\beta$), respectively: 1) air from aloft, originally with concentration ratio $C_E$, that has been recently integrated into the MBL by sinking or mixing, and 2) air that has been in the MBL for considerable time and has become essentially identical in properties to the modeled air with concentration ratio $C$. Thus, $C_C = \beta C_E + (1-\beta)C$ and Eq. (7) can be written in terms of $C_E$ as:

$$\rho \frac{\partial C}{\partial t} = \beta D(C_E - C) - w(z)\frac{\partial C}{\partial z}.$$ 
(8)

## 2.3 Governing Equations

To find the general form of the steady-state equation of conservation of mass for each vapor isotopologue, we combine the diffusive (Eq. 3) and kinematic (Eq. 8) effects and set $\frac{\partial C}{\partial t} = 0$:

$$\rho \frac{d}{dz}\left(K(z)\frac{dC}{dz}\right) + \beta D(C_E - C) - w(z)\frac{dC}{dz} = 0. \tag{9}$$

(Since dynamic variables are used here, this result does not depend on the commonly invoked isopycnal approximation.)

Eq. (9) is the general form of the basic governing equation that we solve in layers in which $K(z)$ and $w(z)$ change. This governing equation is implemented three times, once for each isotopologue, with $K$ differing among isotopologues. Equivalently, it may be viewed as a single vector equation of length 3, with each component describing mass conservation for one isotopologue. The method devised here to solve Eq. (9), described in 3.2, uses the latter strategy.

We now proceed to adapt Eq. (9) to the atmospheric conditions specific to layers 1-3 -- the low, middle and high layers -- of the MBL.

### 2.3.1 Low layer equation

In the low layer, as described at the beginning of this section and as illustrated in Figure 1, there is no convergence or uplift. Hence $D = 0$ and $w(z) = 0$. As specified by Eq. (4), $K$ increases linearly with height, from the small molecular value $K_m$ at the surface ($z = 0$) to the larger mixing rate $K_{max}$ at $z=h_1$, the top of the low (von Kármán) layer, where

$$K_{max} \equiv K_m + b \cdot h_1 \tag{10}$$

In the low layer, Eq. (9) thus simplifies to:

$$b\frac{dC}{dz} + (K_m + b \cdot z)\frac{d^2C}{dz^2} = 0 \tag{11}$$

### 2.3.2 Middle layer equation

In the middle layer, where $h_1 \leq z \leq h_2$, $K(z)$ is constant and equal to $K_{max}$, and the convergence rate $D$ is also constant. Defining $w_a$ as the upward velocity at the top of the middle layer, $h_2$, Eq. (6) implies that $w(z)$ increases linearly from $w=0$ at $z=h_1$ to $w=w_a$ at $z=h_2$. i.e.,

$$w(z) = D \cdot (z - h_1), \text{ and} \tag{12}$$

$$w_a = D \cdot (h_2 - h_1) \tag{13}$$

Eq. (9), after substituting Eqs. (12)-(13), simplifies to:

$$\rho K_{max} \frac{d^2}{dz^2} C + \frac{\beta w_a}{(h_2 - h_1)} (C_E - C) - \frac{w_a(z - h_1)}{(h_2 - h_1)} \frac{dC}{dz} = 0 \tag{14}$$

Within the middle layer, vertical mixing (the first term in Eq. 14) is controlled by the constant eddy diffusion coefficient, $K_{max}$, which is the maximum value of $K$. The second term in Eq. (14) describes the direct effect of convergence of external air from aloft, originally of concentration ratio $C_E$, into the profile. Vertical advection (the third term) occurs at a rate depending on the linearly increasing velocity and the gradient of $C$.

### 2.3.2 High layer equation

The upper layer of the MBL, just below the very top, is often capped by a stable inversion in which diffusion plays a minimal role. Uplift, however, continues upward unabated through the inversion into the free atmosphere, where further evolution of the air mass is beyond the scope of the IMBL model. In the upper layer of the MBL, we assume that $K(z)$ decreases linearly from $K_{max}$ at $z=h_2$ to $K_t$ at the top of the MBL ($z=h_3$), and that there is no further convergence, so $w(z)$ here equals $w_a$. Eq. (9) thus becomes:

$$\rho \frac{d}{dz} \left[ \left( K_{max} - \frac{(K_{max} - K_t)(z - h_2)}{h_3 - h_2} \right) \frac{d}{dz} C(z) \right] - w_a \frac{d}{dz} C(z) = 0 \tag{15}$$

## 3 Solution Methods

### 3.1 Analytic Solutions

All three governing equations, (Eqs. 11, 14 and 15), are second order linear ordinary differential equations with non-constant coefficients. Eqs. (11) and (15) are homogeneous, while Eq. (14) is inhomogeneous by virtue of $C_E$. Each equation has an analytic solution with two constants of integration, totaling six constants requiring six boundary conditions (BC's). The six BC's are:

$C(0)$ is in equilibrium with the surface water.                (BC1)

$C(z)$ and $KdC/dz$ are continuous across $z = h_1$.         (BC2-3)

$C(z)$ and $KdC/dz$ are continuous across $z = h_2$.         (BC4-5)

$dC/dz$=0 at $z = h_3$.                               (BC6)

In the low layer, the solution of Eq. (11) is

$$C(z) = \frac{C_0 ln[h_1 K_{max}]+(C_1-C_0)ln[h_1 K_m + z(K_{max}-K_m)]-C_1 ln[h_1 K_m]}{ln[K_{max}/K_m]} \tag{16}$$

From (BC1), the constant $C_0$ is the isotopologue concentration ratio in equilibrium with the liquid sea surface at the sea surface temperature (SST) (Horita et al., 2008), which we obtain from the specified isotopic composition of ocean water and the fractionation factors between liquid water and vapor (Majoube, 1971). Kinetic fractionation is caused by vertically distributed molecular processes concentrated mostly between the surface and $z=z^*$, and is explicitly included by the presence of $K_m$ in Eq. (16). This treatment of kinetic fractionation, alone, distinguishes between this IMBL and most other models of atmospheric vapor isotopes near the sea surface.

The second constant of integration in Eq. (16) is $C_1$, which is the value of $C(z)$ at $z=h_1$. This constant cannot be evaluated at this point, but we return to it shortly.

Similar to the low layer, the middle and high layers have analytic solutions. As is standard with boundary condition problems, the general solutions are found first. Then BC's 2-5 are introduced into

the solutions, and the four new constants of integration are solved for (in terms of the model parameters). The results are given in Texts S1 and S2 in Supporting Information, respectively.

The solutions given by S1 and S2 are long expressions that are far less amenable to evaluation and interpretation than Eq. (16), their equivalent for the low layer. Furthermore, they still contain constants $C_0$ and $C_1$, introduced from Eq. (16) via the BC's for continuity at $z=h_1$. Thus, the solutions for the middle and high layers cannot be evaluated until after $C_1$ has been found.

In order to find $C_1$, it is necessary to apply (BC6) to equations in S1 and S2 (Supporting Information). The somewhat lengthy result is the equation in S3 in Supporting Information. Once $C_1$ has been evaluated, it is feasible (but tedious and slow) to evaluate equations given in S1 and S2 along with Eq. (16), completing the evaluation of the unique solution set.

## 3.2 Hybrid Analytical/Numerical Solutions

It is more convenient to use a hybrid analytical/numerical approach to finding the solution set. The simple analytic solution for the low layer (Eq. 16) can be evaluated in conjunction with a numerical solution for the middle and high layer equations (Eqs. 14 and 15).

Numerical boundary value problem solvers normally require the specification of boundary conditions containing only the variables, their derivatives, and numerical constants. Such a solver would not be of use, here, because the constant $C_1$ is not known *a priori*, so (BC2) and (BC3) cannot be invoked. However, Matlab's © boundary value problem solver **bvp5c** offers the option of specifying one unknown "parameter" together with two second order boundary value problems and five (instead of the usual four) boundary conditions, and solving for the unknown parameter as well as the continuous variables. In the analytic problem, this would be equivalent to using 5 boundary conditions to solve for four unknown constants of integration and one unknown "parameter" ($C_l$), essentially what was described in Section 3.1.

The Matlab © function **PBL_analy_numer**, in Text S4 of Supporting Information, uses this technique to solve for the isotopologue profiles in the MBL. It calls the solver **bvp5c** (line 143). The

solver **bvp5c**, in turn, calls the function **res** (line 416), for the boundary conditions. Since $C_1$ appears in (BC2), it can be designated by **res** as an "unknown parameter", and the five other boundary conditions (BC2-6) can be specified. The boundary value problem that governs the isotopologue profiles in the MBL is thus completely determined.

## 3.3 Summary

Table 2 contains a list of the eight model parameters that must be specified based on environmental information, and the eight model outputs (either prognostic or diagnostic variables) that are routinely calculated by the model (others can be added). Remember that $C$ is a vector of dimension 3, corresponding to three isotopologues.

## 4 Data for Model Validation

We use seven published data sets for verification of and comparison with our model output. All of these data sets were collected by shipboard measurements. The summary information is included in Table 3, and cruise tracks are illustrated in Figure 2. Samples from these cruises cover a wide range of the world oceans, from the Arctic Ocean to the northern coast of Antarctica. For earlier data sets, i.e., those by Uemura et al. (2008) and Kurita (2011), samples were collected by the cold trap method, and each sample represents an average of 2-12 hours of vapor trapping. Data from the latter five cruises reported by Benetti et al. (2017) were collected by isotope vapor analyzers with the reported instrument model included in Table 3. Benetti et al. (2017) published data sets with either 15 min or 6 hr resolutions; the 6-hour average data are used for this work. The sea surface temperature (SST), which was either directly measured or estimated by the authors, is reported in all datasets. The relative humidity with respect to SST, $RH_{SST}$, is either reported (Benetti, et al., 2017) or can be calculated based on the measured air temperature and relative humidity at the sampling height. Both SST and $RH_{SST}$ are important variables in our model validations.

## 5 Distribution of Parameters for Verification Runs

In this section, we discuss the ranges of parameter values used in the IMBL model verification simulations. The values are summarized in Table 4.

## 5.1 Sea Surface Temperature (SST)

The range of SST used in the simulations was from –2 to +30℃, covering the range of the cruise data sets in Table 3.

## 5.2 Heights $h_{1, 2, 3}$

A finite span of values was not used for either $h_2$ or $h_3$, because results are insensitive to both, and computations were thus reduced in number. The single value used for the MBL height ($h_3$) was 1000 m, a typical MBL height (Stull, 1988), especially in convergent vapor source areas. Similarly, 650 m was the only value used for $h_2$. On the other hand, a full range of values was used for $h_1$, because an informal survey of marine radiosonde data suggests that $h_1$ may range from 50 to 200 m and our results

are sensitive to the value of $h_1$.

## 5.3 Eddy Diffusivity $K_{max}$

The eddy diffusivity, $K$, in the atmosphere boundary layer varies widely over many orders of magnitude. Stull (1988) cited values from 0.1 to 2000 m² s⁻¹, with typical values on the order of 1 to 10 m² s⁻¹ for the atmosphere boundary layer. Olivié et al. (2004) presented a calculated range of 0.01 to

3000 m² s⁻¹ in the lowest 3 km of the atmosphere for 15 days in July, 1993 at two continental and one marine locations; their maximum value ($K_{max}$) above the Pacific ocean location ranged from about 3 to 300 m² s⁻¹. Holtslag and Boville (1993) reported calculated zonal and 31-day average eddy diffusivities between 60°S and 60°N; $K_{max}$ ranged from 20 to 60 m² s⁻¹. For $K_{max}$ greater than 10 m²s⁻¹, model isotopic ratios change only negligibly. At $K_{max}$ values less than 1 m²s⁻¹, the kinetic isotopic fractionation

increases significantly as $K_{max}$ decreases. We, therefore, use a $K_{max}$ range from 0.01 to 100 m²s⁻¹, to obtain the full extent of kinetic fractionation.

## 5.4 Properties of Subsiding Air ($r_E$, $C_E$, $\beta$)

Modeling of convergence requires knowledge of the mixing ratio of the descending air ($r_E$ in g vapor kg⁻¹ dry air) and its isotopic compositions ($C_E$), as well as its proportion ($\beta$) in the air converged into the

MBL. Recall that $C_E$ is a vector of length three, corresponding to the concentration ratios of the three

modeled isotopologues. The $C_E$ value of $H_2^{16}O$ is only very slightly less than $r_E$, while values of $C_E$ for the other two isotopologues can be obtained from $r_E$ and isotopic ratios ($\delta D$ and $\delta^{18}O$) of the vapor.

Vertical profiles of $r_E$ over the ocean have been well observed. We used standard resolution radiosonde data from the University of Wyoming (http://weather.uwyo.edu/upperair/sounding.html) to examine

typical tropospheric values and vertical profiles of the mixing ratio. Generally, the mixing ratio decreases rapidly with height within the lower troposphere, and becomes quite small above mid troposphere. Subsiding air originating in the high troposphere has a correspondingly low mixing ratio. For example, at 500 hPa, the summer-averaged mixing ratio value in the ECMWF (European Centre for Medium-Range Weather Forecasts) data varies from 0.5 to 2 g kg$^{-1}$. Most cruise data in Table 3 were

obtained between summer and fall, particularly the high latitude ones, and we thus use a range of 0.5 – 2 g kg$^{-1}$ for $r_E$ (Table 4) in the simulations.

Measurements of the isotopic composition of vapor are scarce at high altitude. Worden et al. (2007) determined the isotopic composition of tropospheric water vapor from global satellite observations. Values of $\delta D$ averaged over the altitude range corresponding to pressures between 800 and 550 hPa

were found to vary from –180‰ to –250‰ over the extra-tropical ocean. A more recent update reported $\delta D$ values from –140‰ to –250‰ between 900 and 425 hPa (TES$_{v5}$ from Sutanto et al., 2015). Ehhalt's (1974) measurements from aircraft above the Pacific Ocean offshore of Santa Barbara, California showed vertical variations of $\delta D$ from –96‰ to –462‰ between 15 to ~10,000 m for all seasons. The averages for all seasons range from –205 at 800 hPa to –303‰ at 550 hPa. Ehhalt's range is lower, but

overlaps the range of satellite values (Worden et al., 2007; Sutanto et al., 2015). There are no corresponding measurements of $\delta^{18}O$. For the verification simulations, we use a representative value of –239‰ for $\delta D$, and –33‰ for $\delta^{18}O$ (Table 4). Although this choice is somewhat arbitrary, we show that it is adequate for most cruise data sets. To demonstrate the effect of this value, we also show model results with $\delta^{18}O$ of –28‰, as a comparison.

The proportion, $\beta$, of mid-tropospheric air within air converged into the modeled column of the MBL varies with atmospheric conditions including MBL stability, wind speed, and surface roughness. We use a range of values for $\beta$ from 1% to 10%, which are conjectured values, in the verification simulations.

## 5.5 Upward Velocity ($w/\rho$)

NCEP/NCAR reanalysis data (Kalney et al., 1996) show that the upward velocity at 850 hPa ranges globally from 0.01 to 0.4 Pa s$^{-1}$ in magnitude. A typical value of $w$ is 0.1 Pa s$^{-1}$ over the ocean in summer. Higher values of upward velocity can be driven by deep convection, which may, in turn, be
driven by, *e.g.*, long wave radiative cooling at cloud tops. However, the sensitivity of MBL isotopic ratios to $w$ decreases with larger $w$. We thus use a range from 0.012 to 0.18 Pa s$^{-1}$, corresponding to 0.01 to 0.15 m s$^{-1}$.

## 5.6 Other Parameters and Constants

In addition to the parameters discussed above, a few more parameters and/or constants are needed for
the simulations. For the isotopic compositions of ocean water, both $\delta D$ and $\delta^{18}O$ are set to zero. The molecular diffusivity of $H_2^{16}O$ in air is assumed to equal that of bulk water vapor, whose temperature dependence in m$^2$s$^{-1}$ is given by the polynomial fit to Bolz and Tuve's (1976) data (Nellis and Klein, 2009), $K_m = -2.775E\text{-}6 + 4.479E\text{-}8*SST + 1.656E\text{-}10*SST^2$. The molecular diffusivities of $H_2^{18}O$ and $HD^{16}O$ are both smaller than that of $H_2^{16}O$ by factors of 0.9723 and 0.9755, respectively, based on
values of Merlivat (1978). The turbulent diffusivity at the top of the MBL is set to 100 $K_m$; while there is little data with which to justify this choice, it suffices because the results are insensitive to it.

The values listed in Table 4 yield 2835 combinations, the result of which is the set of model results we discuss in the next section.

## 6 Results and Discussion

In this section, we discuss the characteristics of the model output and their physical significance, and compare the output with observations. We first show vertical profiles of isotopic properties of vapor in the MBL for a representative set of parameters, and then we present the entire set of result of 2835 calculations. These results are then compared with cruise data in both $\delta D$ vs. $\delta^{18}O$ and d-excess vs. $\delta^{18}O$ spaces. We end the section with a discussion of model limitations and potential future developments.

## 6.1 Characteristics of Model Results and Model Validation

While a full discussion of parameter sensitivities and the associated physical processes is the subject of an anticipated companion paper, we point out a few major features of the model output that will guide our discussion of model validation. We start by presenting vertical profiles of δD, $δ^{18}O$ and d-excess. We do so to emphasize that this model is a true 1D model, unlike Craig-Gordon type models. We also emphasize the points that 1) there are strong gradients near the air-sea interface, and 2) all isotopic vapor observations made during marine research cruises are done at a single height, corresponding to just one point of each of the δD, $δ^{18}O$ and d-excess profiles. We then discuss the δD-$δ^{18}O$ and d-excess-$δ^{18}O$ relationships, which are of major importance to the isotopic interpretation of vapor and precipitation (both modern and ancient such as tree rings and ice cores).

### 6.1.1 Vertical profiles

As a one-dimensional model, the IMBL model yields the vertical distribution of the isotopic quantities δD, $δ^{18}O$, and d-excess ($=δD–8δ^{18}O$). Figure 3 illustrates a typical result. Vapor isotopic values $δ^{18}O$ and δD are both high near the sea surface, where vapor is in equilibrium with ocean water. With increasing height, isotopic ratios and humidity decrease because of the mixing of MBL vapor with isotopically depleted vapor that descends from the upper atmosphere outside, and then is converged into, the modeled column. The upper atmosphere vapor has much lower values of both δD (–239‰) and $δ^{18}O$ (–33‰), but a higher value of d-excess (25‰), than vapor in equilibrium with ocean water.

The profiles in Figure 3 display strong curvature with very steep gradients near the sea surface, diminishing to negligibly small gradients throughout the MBL. This curvature arises from the rapid change of *K* from very small molecular values within the thin laminar layer near the water-air interface to large turbulent values above the laminar layer. In this work, the thickness of this layer is characterized by $z^*$, the height of the crossover between molecular and turbulent diffusivities, below which turbulent diffusion is suppressed (See Eqs. 4 and 5).

The small molecular diffusivity that dominates diffusion in the laminar layer -- in particular, its differences among isotopologues -- is the cause of kinetic fractionation. Kinetic isotope fractionation is

reflected by d-excess that changes more sharply near the surface than does either $\delta D$ or $\delta^{18}O$. The smaller inset of d-excess vs. height plot shows variations within 20 cm of the water-air interface. The $z^*$ value, which is 2.7 cm in this particular run, is indicated in the inset by the dashed line. The effect of turbulent diffusion increases with height, and thus the rate of change in d-excess with height decreases

rapidly as the height increases.

Most in situ observations are conducted at a fixed height above the sea surface. The seven cruise data sets (Table 3) were collected at heights between 10 and 20 m. In these cases, each measurement represents an air sample at a given height along a vertical profile. As shown in the calculation depicted in Figure 3, isotopic gradients are greatest near the sea surface; in this example, over just 15 m (which is

only 1.5% of the total height of the MBL) $\delta^{18}O$, $\delta D$ and d-excess achieve 58, 43, and 88%, respectively, of the change toward the relatively constant values between $h_2$ and $h_3$ (650-1000 m). Above 10 m, isotopic change with height is relatively slow. For example, in this particular calculation, at 15 m the $\delta^{18}O$, $\delta D$, and d-excess values are –15.6, –112.6‰, and 12.2‰, respectively; they change by only 0.50, 3.56, and 0.40‰, respectively, between 10 and 20 m. Consequently, the isotopic variations between 10

to 20 m to be discussed in the upcoming sections can be viewed as approximating the isotopic variations of vapor delivered to the free troposphere. If the actual vapor isotopic ratios of an air mass to initiate a Rayleigh process are desired, the values at $h_3$ should be used.

### 6.1.2 The $\delta D$ vs. $\delta^{18}O$ relationship

Each of the 2835 combinations of parameter values described previously was used for one model run.

Isotopic ratios were calculated at 15 m above the sea surface and plotted in $\delta D$-$\delta^{18}O$ space (Figure 4, main graph). The choice of 15 m height for Figure 4 is somewhat arbitrary, but is approximately the average of the observation heights, that range from 10 to 20 m, in the seven data sets with which we compare our results (Table 3). In the upper small inset, superimposed in red over the 15 m values are isotopic ratios at both 0 m (in equilibrium with seawater at 5°C) and 15 m for the particular simulation

presented in Figure 3, giving a different perspective on the vertical isotopic change. Vapor at 15 m for this particular run has about average deviation from the sea surface equilibrium vapor. Other runs may have larger or smaller vertical gradients in either or both $\delta D$ or $\delta^{18}O$. The magnitude of the vertical

gradient is reflected by the value of $z^*$. Among the 2835 runs, the distribution of $z^*$ is right skewed with a range from 0.001 to 52 cm and a median of 2.8 cm. This median $z^*$ value is similar to and thus well represented by the particular run in Figure 3 ($z^*$=2.7 cm). As discussed earlier, most changes occur below 10 m; above 10 m the change in isotopic composition is relatively minor.

The lower small inset in Figure 4 shows a comparison of two sets of simulations (2835 runs each) using different oxygen isotopic ratios for the upper atmosphere air. Only the boundaries of the output areas are shown, with blue being identical to the main graph, and red indicating the range of results produced using –28‰ (rather than –33‰) for the $\delta^{18}O$ value of the upper atmosphere vapor.

The output in $\delta D$-$\delta^{18}O$ space (Figure 4) defines a quadrilateral with each corner labeled A through D.
The edges (BC, CD, DA and AB) have specific physical significance. Line BC (line **b**) connects all points (squares) representing isotopic values of vapor in equilibrium with seawater, for the range of sea surface temperatures considered. With increasing sea surface temperature, the points shift from lower left (C) to upper right (B). Points close to this line reflect model parameters that permit very little kinetic isotopic fractionation to occur between the sea surface and 15 m, and very little influence of
descending air (whose isotopic composition is point E). Close examination reveals that the points near line BC were generated with the largest turbulent mixing coefficients (highest $K_{max}$), and a very small fraction of external air (small $\beta$~0.01). Consequently, $z^*$ values are very small (~1x10$^{-5}$ m), and the relative humidity with respect to SST, RH$_{SST}$, is close to saturation, both of which are responsible for the small degree of kinetic isotopic fractionation. Large $K_{max}$ also creates well-mixed MBL, which is
consistent with the simulated low isotopic gradients between the sea surface and 15 m.

Line CD bounds points that have the smallest deviation from the straight line CE (line **c**) that represents mixing of vapor in equilibrium with SST at the coldest temperature considered (–2°C, point C) and vapor from the descending high altitude dry air (E). Increasing contribution from kinetic isotopic fractionation moves points increasingly above this line (see further discussion below). Therefore, points
on this line represent no kinetic fractionation, with the influence of upper atmosphere air increasing from C to E. In other words, if the SST is –2°C, line CE represents a lower bound on isotopic mixing. At a fixed SST and ocean isotopic ratio, this line rotates with changing isotopic ratios in the air aloft, for

example, line CF in the lower inset of Figure 4. Similarly, mixing lines between equilibrium vapor at higher SST's should be straight lines connecting point E and points along line $b$ representing vapor in equilibrium with seawater at different temperatures. For example, if the SST is 30°C, then the mixing line would be BE (not shown), and all isotopic ratios of vapor evaporated from this sea surface should be above this line.

The points along line AB represent vapor evaporated from ocean water at SST=30°C. Their spread reflects the influence of kinetic fractionation; moreover, they are not significantly influenced by mixing with upper atmosphere air. This inference is supported by their small values of $K_{max}$ (0.1 m$^2$ s$^{-1}$), large $z^*$ (0.3-0.5 m), and low $r_E$ of the upper atmosphere (0.5 g kg$^{-1}$). Theoretically, the slope of AB should have a limit of 0.88 (shown as line $a$), the ratio of the kinetic fractionation factors of HDO and H$_2{}^{18}$O (25.1 and 28.5‰, respectively, because $K_m/K^*_m$=1.0251 and 1.0285, respectively, where the star represents the heavy isotopologue; Merlivat, 1978). With the set of parameters in Table 4, the slope of AB is about 1.5, slightly greater than its lower limit (0.88). Therefore, line $a$ sets the upper bound for vapor isotopic ratios for SST of 30°C. In other words, the theoretical limit for the highest isotopic ratios at a given SST should be along a line that starts from a point representing vapor in equilibrium with seawater ($\delta^{18}$O=0, $\delta$D=0 in this calculation) at this temperature and extends to the lower left with a slope no less than 0.88.

Line AD bounds isotopic compositions reflecting the entire range of SST values; both kinetic fractionation and mixing with the upper atmosphere have significant influences on these points. The ambient conditions are characterized by small $K_{max}$ (0.01 m$^2$ s$^{-1}$), large $\beta$ (0.1) and relatively high $z^*$ values (0.1-0.5 m). This AD "line" is not as strictly defined as other lines in that it does not have an absolute theoretical limit and so may change with the range of parameter space. In subsequent discussion, we refer to line $a$ as the upper limit, line $b$ the side limit, and line $c$ the bottom limit of the vapor distribution in the $\delta$D-$\delta^{18}$O space, consistent with their positions in Figure 4.

In summary, the shape of the output in Figure 4 is controlled by three factors, 1) the SST, 2) the degree of kinetic isotopic fractionation, and 3) the influence of upper atmosphere air. While SST is relatively independent of other factors, kinetic fractionation and effect of upper air depend on various

combinations of atmospheric conditions, including the intensity of turbulent mixing ($K_{max}$), the mixing ratio of the descending air ($r_E$) and its isotopic ratios, the proportion of upper atmospheric air advected into the evaporation column ($\beta$), and the vertical velocity ($w$). Note that in this model, the relative humidity with respect to SST ($RH_{SST}$) is not, and cannot be, prescribed. On the contrary, it is an outcome of the same meteorological conditions of the MBL that affect the isotopic ratios, although it also feeds back on kinetic isotopic fractionation by controlling the vertical gradient for vapor diffusion.

Model output and observational data for each individual cruise are compared in Figure 5. Model output is calculated at the observation height of the corresponding cruise, indicated in the graph. Also included in each plot are compositions of vapor in equilibrium with seawater at the lowest and highest SSTs of the cruise. The theoretical borders under specific cruise conditions are shown as solid lines; observed isotopic ratios are expected to fall within these theoretical limits (if consistent with the assumed ocean water and descending air isotopic ratios).

We make the following observations of Figure 5. First, the vast majority of the observed data (~95%) do indeed fall within the expected range. This confirms not only the successful conceptualization of the model but also that our choices of parameter values are reasonable.

Second, in all seven data sets, the influence of the isotopically depleted vapor from descending air is demonstrated by points with low isotopic ratios. These compositions are difficult to model using C-G type models, particularly with the invalid closure assumption (e.g., Jouzel and Koster, 1996; Benetti et al. 2015). This result highlights the importance of convergence in affecting boundary layer vapor isotopic ratios, as it introduces dry, depleted air from aloft into the MBL. Such influence of upper atmosphere air on the boundary layer has been recognized by Benetti et al., (2015, 2018), although for quiescent subsidence regions that our model does not treat.

Third, for the ACTIV cruise (Figure 5c), a number of points fall below the lower limit, suggesting that the isotopic ratios of the descending air we used for the simulation may not be representative in this area during the observation period. The mismatch suggests a value that is more enriched in $^{18}O$, or depleted in deuterium, or both, than the values used for the simulation.

Fourth, in four cruises (PIRATA, STRASSE, Bermuda and RARA; Figures 5d-g), there are points that are above the upper limit. However, in all cases except RARA, the enrichment above the upper limit is small in magnitude, and may be explained by slight variations in seawater isotopic ratios. For RARA, however, the enrichment above the upper limit is significant. One possible explanation is the influence

of sea spray. When describing the sampling conditions, Benetti et al. (2017) particularly noted that they could not completely rule out the contribution of small droplets of sea spray to the vapor composition. However, such an influence seems relatively small considering the great leverage of seawater isotopic composition. Figure 5h shows the direction and magnitude of sea spray influence; the mixing of sea spray droplets should cause enrichment such that the data would be distributed in the triangular area

bordered by the dashed lines. Detailed examination of cruise logs in the future will be helpful to confirm and quantify the sea spray contribution to MBL vapor.

In summary, by comparing calculated values and observational data in $\delta D$-$\delta^{18}O$ space, we conclude that the model is remarkably successful. We pointed out three factors that may cause observations to fall outside the predicted range, namely 1) variation in ocean water isotopic ratios, 2) variation in the

isotopic ratios of the upper atmospheric vapor, and 3) influence of sea spray on vapor isotopes. In section 6.2, we discuss several other model assumptions that may limit the consistency between model results and observations.

### 6.1.3 Deuterium excess (d-excess)

The relationships between d-excess and both sea surface temperature (SST) and relative humidity with

respect to SST (RH$_{SST}$) have been intensively discussed by the isotope hydrology community. Originally defined by Dansgaard (1964) for precipitation as $\delta D–8\delta^{18}O$, d-excess has been used to infer conditions at the moisture source location. It has been demonstrated that d-excess varies with SST and inversely with RH$_{SST}$ (Johnsen et al, 1989; Petit et al., 1991). Later investigators have used these concepts to infer changes in moisture source conditions recorded in polar ice cores (e.g., Vimeux et al.,

1999; Masson-Delmotte et al., 2005a, 2005b).

The relationships between d-excess and SST, and between d-excess and RH$_{SST}$, are shown in Figure 6. Our model, as expected, exhibits a significant dependence of d-excess on both SST and RH$_{SST}$.

Regression of d-excess against SST explains 16% of the variance in d-excess, with a coefficient of 0.35‰ $°C^{-1}$. Regression against $RH_{SST}$ explains 78% of the variance in d-excess, with a coefficient of –0.43‰ $\%^{-1}$. These values are very similar to d-excess sensitivities of 0.35‰ $°C^{-1}$ to SST and –0.45‰ $\%^{-1}$ to $RH_{SST}$, respectively, cited by Vimeux et al., (1999) based on calculations by Johnsen et al.,

5   (1989).

All three processes discussed earlier, i.e., changing SST, degree of kinetic fractionation, and extent of mixing with the subsiding air, result in changes in d-excess. This is seen by the fact that the theoretical lines in Figure 6 representing each of the three processes have non-zero slopes. Although at the sea surface d-excess increases with SST, a much larger spread occurs at 15 m due to the height-dependent

influence of the descending air and kinetic fractionation. For each value of SST, the points at 15 m form a triangular area, within which d-excess varies significantly (Figure 6a). Such a triangular distribution of isotopic data in the d-excess vs. $\delta^{18}O$ space has been reported by Steen-Larsen et al. (2015) for observations off the coast of Iceland. This two-dimensional distribution explains the significant, though relatively poor, correlation between d-excess and SST.

Figure 6b shows that the lowest $RH_{SST}$ tends to correspond to the highest d-excess distributed near corner A. However, near point D, where d-excess is also relatively high compared with values at the water-air interface, $RH_{SST}$ is relatively high and kinetic fractionation is limited. Therefore, while d-excess tends to increase as $RH_{SST}$ decreases, the relationship is not one to one (note how color changes horizontally in Figure 6b). Another way to see this is to trace the color change along lines parallel to CB

in Figure 6a for changes in SST, and parallel to AB in Figure 6b for changes in $RH_{SST}$. Interestingly, along CD, neither SST nor $RH_{SST}$ varies significantly, regardless of substantial variation in d-excess. Obviously, $RH_{SST}$ is not the sole influence on d-excess, and even the combination of both $RH_{SST}$ and SST does not completely determine d-excess in the MBL.

Data from all cruises are shown in Figures 6c and d. In order to pool a larger quantity of data, we ignore

here the minor differences in sampling heights among the seven data sets. The match between observed and simulated patterns is excellent. First, ~95% of data fall within the theoretically predicted region (the percentage may be slightly less than 95%, because the simulation here is done only at 15 m without

considering the sampling height of each cruise). This comes as no surprise given what was already seen in $\delta D$-$\delta^{18}O$ space (Figure 5). Factors that cause a small number of observational points to fall outside the predicted region were discussed earlier, and we do not repeat that discussion here. Second, the dependence of the observed d-excess on $RH_{SST}$ and SST, as shown by the color distributions, is very similar to that of model calculations. For SST, the color along lines parallel to CB changes from green to red with a d-excess increase. Similarly, $RH_{SST}$ values are relatively high near lines CB and CE, and decrease (with significant noise) towards corner A as d-excess increases. Finally, as predicted, the observed d-excess correlates significantly (p<0.0001) with SST and with $RH_{SST}$. The sensitivity of d-excess observations to $RH_{SST}$ is –0.36‰ %$^{-1}$, comparing favorably with corresponding model sensitivity of –0.43‰ %$^{-1}$. For SST, the sensitivity for observations is 0.15‰ $^o$C$^{-1}$, about half of that predicted by the IMBL model (0.35 ‰ $^o$C$^{-1}$) using SSTs ranging from –2 to 30$^o$C. Detailed discussion of the sensitivity differences between simulations and observations is beyond the scope of this paper, and a full understanding of these relationships should be an important goal for future investigations.

To end this section, we point out that our model-observation comparisons are focused on identifying major processes controlling large-scale isotopic distributions of water vapor. These general comparisons should be followed by simulations specific to given sites over given observation time windows, which would require narrowing down model parameterizations according to the conditions where and when data were collected. For example, the SST, water isotopic values, vertical velocity, $K_{max}$, properties of descending air should all be obtained/estimated either from observations or from reanalysis products. Such site- and time-specific simulations will allow identifying relative importance of various processes and will lead to an understanding of how the relative contributions of each process vary over space and time. Since such work requires a particular context for each data set, we postpone it to future investigations.

**6.2 Model Applicability, Limitations and Future Development**

The IMBL model described here has considerable potential value for many isotope hydrology applications. First, as vapor isotopic measurements become increasingly available, application of the model at different locations and times of year provides a vehicle to explore and understand relationships

between meteorological conditions and isotopic compositions of vapor both within the MBL and delivered to the free atmosphere. Comparisons of simulations and observations with much more intensive observations than cited in this work may be conducted. For example, during an isotopic vapor measurement campaign, measured variations of the isotopic composition of ocean water and vapor may
be combined with model calculations to constrain the diffusion coefficient, which may then be related to sea surface roughness, wind speed, vertical velocity, and sea spray occurrence. Second, the output of this model, i.e., the isotopic ratio of vapor delivered to the free atmosphere, can be used to provide initial conditions for Rayleigh-type condensation or transport models. The sensitivity of precipitation isotopic ratios to the range of meteorological conditions at the moisture source region and their change
over time and space can be investigated for modern hydrological cycles in association with atmospheric circulation, as well as under ancient climate conditions. Third, an understanding of physical processes important for controlling the isotopic compositions of water (both vapor and precipitation), gained from these applications, can be used to improve the physical representation of marine boundary layer processes in GCMs.

This IMBL model may not adequately describe several meteorological scenarios, and its use under those conditions should be made with caution. First, the model requires that the air column in the model domain have a positive (upward) vertical velocity, i.e., air must be converging and rising. This assumption is made to ensure that the model column represents a moisture source area, delivering vapor to the free troposphere where it will ultimately produce precipitation. If the vertical velocity is negative
(i.e., subsiding air), the air in the column diverges rather than converges, meaning that the mass conservation equations used here would not be correct. However, a modest formulation of the governing equations could allow for sinking air (Welp et al., 2018). Either way, the important outcome is that upper atmosphere vapor is incorporated into the MBL. It is possible that isotopic distribution changes somewhat with specific mixing scenario, while the theoretical limits of isotopic distribution
remain the same as shown by this work. Second, the model does not include exchange between vapor and liquid if air is supersaturated, forming clouds or precipitating. The model is thus strictly applicable only for meteorological conditions with no cloud base below $h_3$, the top of the MBL (1000 m, here). Third, as discussed earlier, the model does not include the effects of sea spray. Fourth, the modeled

column is not subject to horizontal advection (except for convergence). Fifth, the IMBL is a steady state model.

We envision future improvements in the IMBL model. In particular, we anticipate generalization of the model to include areas of divergence (descending air), areas with sea spray, and/or terrestrial areas. Alternatively, to describe or simulate effects or dynamics of additional boundary layer processes, such as cloud microphysics, researchers could consider using more complex and comprehensive boundary layer models, such as the ISOLESC model (Wei et al., 2018), with associated sacrifice of simplicity.

## 7 Conclusions

We have constructed a new isotope marine boundary layer (IMBL) model to calculate the isotopic composition of vapor in the marine boundary layer as well as that of vapor lifted to the free atmosphere. The model divides the MBL into three layers, each with its own transport characteristics. Compared with classical Craig and Gordon (1965) type bulk evaporation models, this 1D (vertical) model makes two improvements. First, it explicitly includes the process of horizontal convergence in the middle layer; convergence provides mass to balance the upward advection supplying moisture for cloud formation and precipitation in the free atmosphere. This formulation requires specification of the properties of subsiding air that mixes with low altitude air and converges into the model column. Second the eddy diffusion coefficient is height-dependent, equal to the molecular diffusion coefficients for each isotopologues at the air-water interface, and increasing linearly through the lower layer to a maximum value, $K_{max}$, remaining constant through the middle layer, and decreasing linearly to $K_t$ over the top layer. The advantages gained from these two improvements include 1) the model solves for both isotopologue concentrations in and fluxes through the MBL, not just fluxes; 2) kinetic isotopic fractionation becomes a diagnostic variable rather than a required parameter, without adding to the total number of parameters (degrees of freedom) of the model; 3) relative humidity is also no longer a specified parameter, but rather becomes a diagnostic variable, thus enabling the use of the model to investigate and possibly predict evaporation rates; 4) calculation of vertical profiles of concentrations and fluxes of isotopologues (or isotopic ratios), providing an opportunity to compare model output with observations at a specific height or multiple heights; and 5) the air at the top of the MBL (at $z = h_3$) is

the air mass supplied to the beginning of a Rayleigh trajectory, which can be used for many isotope studies.

The standard output of the model includes vertical profiles of $\delta D$, $\delta^{18}O$ and d-excess. Near the sea surface, $\delta D$ and $\delta^{18}O$ are high and close to equilibrium with the ocean water, and d-excess is low. With

increasing altitude, $\delta D$ and $\delta^{18}O$ decrease due to both kinetic fractionation and mixing with converging isotopically depleted air that contains subsided air from the free troposphere. Kinetic fractionation near the sea surface causes d-excess to increase rapidly with height, particularly between the air-sea interface and height $z^*$, where molecular diffusion dominates over turbulent mixing.

Model simulations using reasonable ranges of parameters are validated using seven sets of shipboard

MBL observations. The resulting range of $\delta D$ and $\delta^{18}O$ forms a quadrilateral-shaped pattern in the $\delta D$-$\delta^{18}O$ plane. Three processes generate boundaries for the quadrilateral, or constraints on the isotopic ratio distributions, including 1) the set of vapor isotopic ratios in equilibrium with seawater at various SST's (right side boundary), 2) mixing between vapor descended from the upper atmosphere and vapor in equilibrium with seawater at the air-water interface (lower boundary), and 3) kinetic isotopic

fractionation that starts with equilibrium vapor and extends to more depleted values of $\delta D$ and $\delta^{18}O$, with a slope no less than 0.88 (upper boundary).

About 95% of observations fall into the theoretically predicted quadrilateral region, demonstrating the success of the model conceptualization and parameter choices. This remarkable agreement highlights the importance of convergence and entrainment of descending, isotopically depleted air to boundary

layer isotopic ratios. This feature is new to this model, and was not considered in earlier simple models, although some (e.g., Benetti, 2015) do include mixing by mathematically unspecified mechanisms other than convergence, in meteorological regions distinct from those we consider. The simulation-observation comparisons also point to at least three factors that may explain the 5% of data falling outside the predicted region, including 1) variations in seawater isotopic ratios, 2) inaccurate choice of

isotopic ratios for the subsiding air, and 3) influence of sea spray. It is also possible that meteorological scenarios not explicitly considered by the model are responsible for the relatively minor model-data mismatch. Such factors may include low level air divergence (downward vertical velocity in the middle

and upper MBL), vapor-liquid exchange (during precipitation events or within clouds), and the presence of lateral advection.

Simulated d-excess compares remarkably well with observations. While the effects of sea surface temperature (SST) and relative humidity with respect to SST ($RH_{SST}$) are well-understood, we draw
attention to the influence of the upper atmosphere air in controlling d-excess in the marine boundary layer. In this simulation, the d-excess value of the descending air is greater than that of vapor in equilibrium with seawater, and the contribution of the former to MBL air results in an increase in d-excess of vapor, even in the absence of kinetic isotopic fractionation. The influence of free troposphere vapor on the d-excess of the boundary layer vapor has also been demonstrated by Benetti et al. (2015,
2018) via a C-G type approach. Quantification of this influence under various meteorological scenarios should be an important objective for future investigation in order to use d-excess for ice core studies.

The IMBL model can be used in a number of ways. First, numerical experiments with the model help to better understand the effects of boundary layer processes and climatic conditions on isotopic compositions of vapor within and vapor fluxes through the MBL. For example, one may
investigate how boundary layer stability, turbulence conditions, vertical velocity, convergence, and upper atmospheric moisture affect MBL isotopic distributions and how these effects change with space and time. A second application could be to investigate how temporal and spatial meteorological differences in moisture source regions affect the isotopic composition of remote precipitation under both modern as well as paleo-climate conditions. In this application, the
IMBL model can be coupled with a Rayleigh distillation model to simulate isotopic evolution of vapor and/or precipitation from moisture source to a precipitation site. These simulations can be particularly powerful if also used in conjunction with a Lagrangian back trajectory programs to identify moisture source areas for a site of interest. Third, it is important to investigate the relationship between MBL isotopes and evaporation rate and, perhaps, to develop methods to
measure the latter indirectly from simultaneous observations of isotopes and meteorological conditions. Since this IMBL model calculates the flux of each isotopologue (rather than just their ratios), it yields the evaporation rate. This opens up the possibility of using isotopic measurements to quantify evaporation rates under various boundary layer conditions. Finally, the

understanding gained from IMBL model simulations can be used to improve the representation of MBL processes in isotope-enabled GCMs.

## Author Contribution

E.S. Posmentier created and coded the IMBL model, wrote the model description of the manuscript and edited all other sections. L. Sonder helped with coding of the model, and did intensive editing of the manuscript. X. Feng did most of the calculations, interpretation of the model output, and drafting of the manuscript. N. Fan researched model parameters, did preliminary sensitivity test of the model, and drafted the descriptions of the parameter ranges, and organized part of the data for model validation. All authors were involved in discussions and interpretations of the model results.

## Acknowledgements

Funding of this work was in part provided by the National Science Foundation under grant 1022032 for the iisPACS (Isotopic Investigation of Sea ice and Precipitation in the Arctic Climate System) project. We thank R. Uemura (personal communication), Kurita et al. (2011) and Benetti et al. (2017) for making their observational data available. Technical assistance by J. Chipman is appreciated. We thank two anonymous reviewers for their thoughtful comments that helped us improve this manuscript. NCEP Reanalysis data were provided by the NOAA/OAR/ESRL PSD, Boulder Colorado, USA from the Web site at https://www.esrl.noaa.gov/psd/.

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

# Table 1. Symbols Used

| Symbol | Description | Units |
|---|---|---|
| $b$ | rate of increase of $K$ with height | length time$^{-1}$ |
| $C$ | concentration ratio of an isotopologue | isotopologue-mass dry-air-mass$^{-1}$ |
| $D$ | horizontal dynamic "convergence" | dry-air-mass-volume time$^{-1}$ |
| $h_{1,2,3}$ | height (z) at the top of the low, middle, and high layers, respectively | length |
| $F$ | vertical flux of an isotopologue | isotopologue-mass area$^{-1}$ time$^{-1}$ |
| $K$ | kinematic diffusion (mixing) coefficient | length$^2$ time$^{-1}$ |
| $K_m$ | $K$ for molecular process | length$^2$ time$^{-1}$ |
| $K_{max}$ | value of $K$ in middle layer | length$^2$ time$^{-1}$ |
| $r$ | mixing ratio | total-vapor-mass dry-air-mass$^{-1}$ |
| $w$ | dynamic vertical "velocity" (kinematic (convectional) velocity (length time$^{-1}$) = $w/\rho$) | dry-air-mass area$^{-1}$ time$^{-1}$ |
| $z$ | vertical coordinate | length |
| $z^*$ | laminar layer thickness scale | length |
| $\beta$ | mass fraction of air from aloft entrained into the MBL. | dimensionless |
| $\rho$ | density of air | dry-air-mass volume$^{-1}$ |
| $u$ | wind speed | length time$^{-1}$ |

**Table 2: Model parameters, results and diagnostics**

| A. model parameters whose values must be specified |
| --- |
| SST, $h_{1,2,3}$, $K_{max}$, $\beta$, $w_a$, $C_E$ |
| B. model results (calculated variables) |
| $C(z)$, $r(z)$, and $F(z)$ *at z=0 and z=$h_3$;  $z^*$*, $E$ (evaporation rate) |

5 **Table 3. Source of data sets used for validation of the IMBL model**

| Cruise | Ship Name | Time Period (dd.mm yyyy) | Method | Height above sea surface (m) | Measure-ment interval (hr) | Reference |
| --- | --- | --- | --- | --- | --- | --- |
| Southern Ocean | R/V Umitaka-maru | 30.12 2005-30.01 2006 | Cold trap | 15 | 2-12 | Uemura, et al., 2008 |
| Arctic Ocean | R/V Mirai | 01.09-05.10 2008 | Cold trap | 20 | 6 | Kurita, 2011 |
| STRASSE | R/V Thalassa | 16.08-13.09 2012 | Picarro L2130i | 17 | 6 | Benetti et al., 2017 |
| PIRATA FR 24 | R/V Suroit | 01.05-20.05 2014 | Picarro L2130i | 12 | 6 | Benetti et al., 2017 |
| RARA | S/V Rara Avis | 26.01-11.06 2015 | Picarro L2120i | 10 | 6 | Benetti et al., 2017 |
| ACTIV | S/V Activ | 23.06-20.09 2014 | Picarro L1102-i | 15 | 6 | Benetti et al., 2017 |
| Bermuda | R/V Atlantic Explorer | 26.09-11.10 2014 | Picarro L2120i | 11 | 6 | Benetti et al., 2017 |

**Table 4. Parameter values used in model verification simulations**

| Parameters | Values | Units |
|---|---|---|
| Sea surface temperature (SST) | -2, 5, 10, 15, 20, 25, 30 | °C |
| Turbulent diffusion coefficient ($K_{max}$) | 0.01, 0.1, 1, 10, 100 | $m^2s^{-1}$ |
| Upward velocity ($w/\rho$) | 0.01, 0.08, 0.15 | $ms^{-1}$ |
| Mixing ratio of subsiding air ($r_E$) | 0.5, 1.2, 2 | $gkg^{-1}$ |
| Fraction of subsiding air ($\beta$) | 0.01, 0.05, 0.1 | |
| Thickness of lowest layer ($h_1$) | 50, 120, 200 | m |
| Upper boundary of middle layer ($h_2$) | 650 | m |
| Height of MBL ($h_3$) | 1000 | m |
| $\delta D$ and $\delta^{18}O$ of subsiding air | -239 and -33, (-28) | ‰ |
| $\delta D$ and $\delta^{18}O$ of ocean water | 0 and 0 | ‰ |

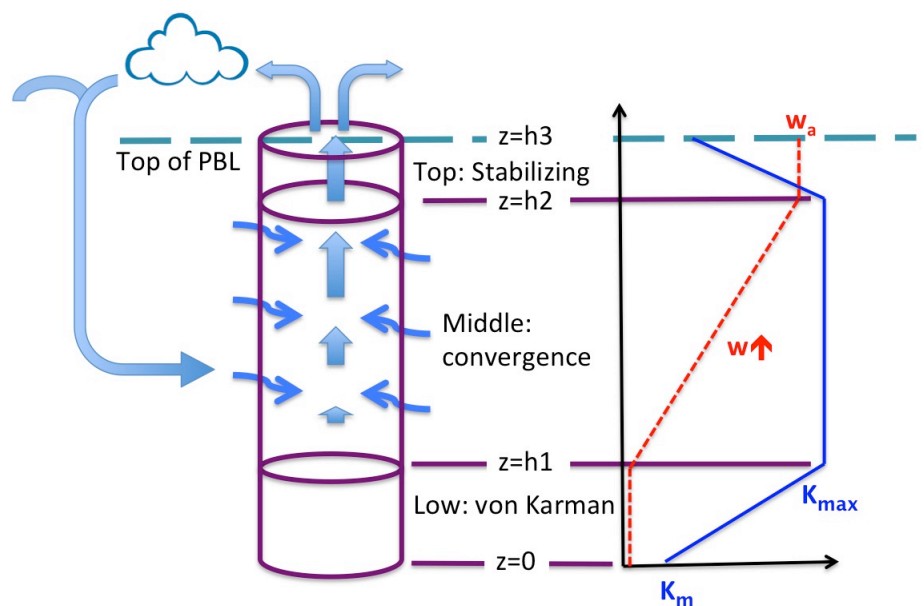

**Figure 1: Cartoon of the MBL model. The modeled region is indicated by the cylinder. It is comprised of three layers – the low von Kármán layer, the middle convergence layer, and the top stabilizing layer. The heavy straight arrows represent the flow of semi-moist air ascending through the middle and top layers, and through the top of the MBL into the free atmosphere (above the model) where clouds and precipitation may form, after which some depleted air from the model column or elsewhere sinks and mixes into the MBL and converges into the middle layer of the model (thin wiggly arrows). Vapor is advected by the vertical motion of air in the middle and top layers and is transported by vertical (diffusive) mixing in all three layers. To the right are graphs of $w(z)$ (dashed red) and $K(z)$ (solid blue) as they vary through the three layers.**

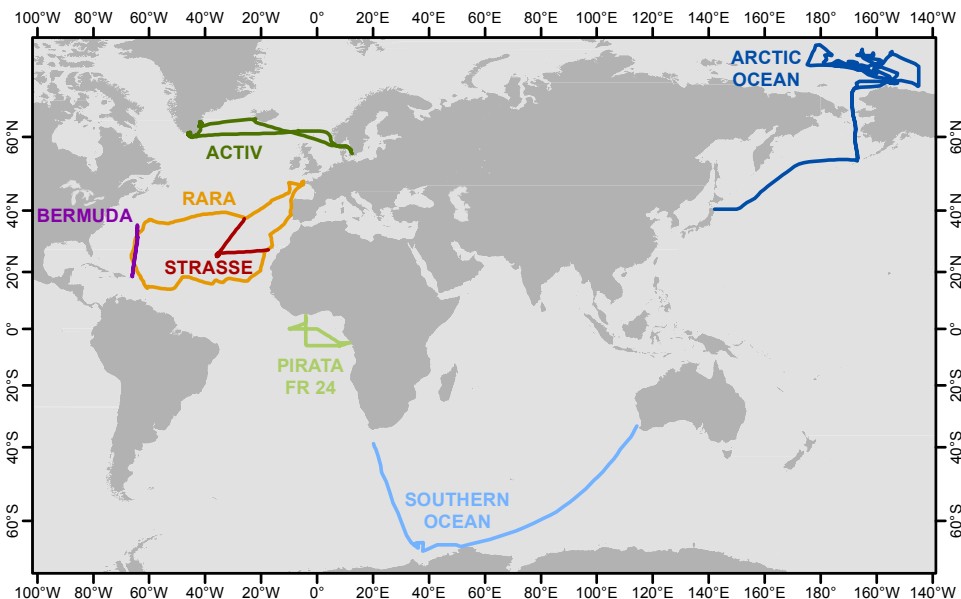

**Figure 2: Map showing the tracks for seven cruises that generated vapor isotopic observations used in this work. Information about each cruise is listed in Table 3 and the associated references.**

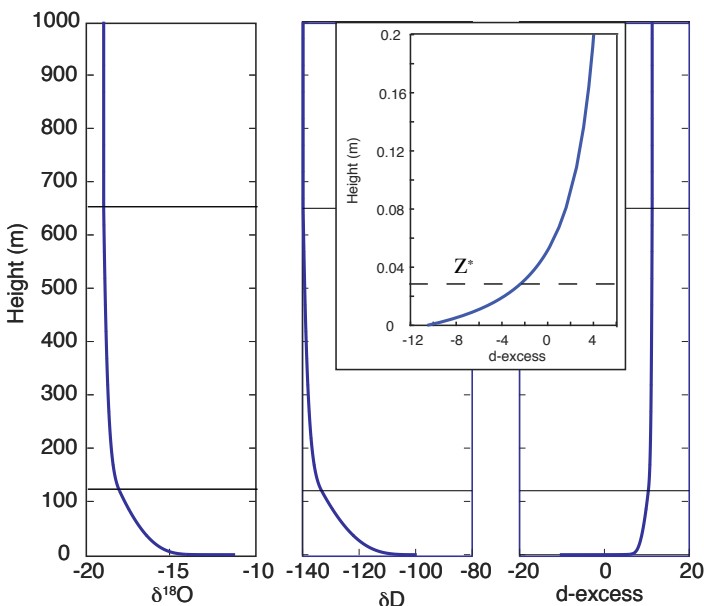

**Figure 3: Model simulation of vertical distributions of $\delta^{18}O$ (left), $\delta D$ (middle) and d-excess (right). Parameters are SST=5°C, $K_{max}$=0.1 m²s⁻¹, $h_1$=120 m, $r_E$=0.5 g kg⁻¹, $w_a$=0.15 m s⁻¹, $\beta$=0.05, and $\delta^{18}O$, $\delta D$ and d-excess of subsiding air are –33, –239 and 25‰, respectively. The horizontal solid lines mark the heights of $h_1$ and $h_2$ (120 and 650 m). The inset graph shows d-excess variation with height in the 20 cm just above the sea surface. The dashed line marks the value of $z^*$ (0.027 m), below which molecular diffusion is more significant than turbulent diffusion.**

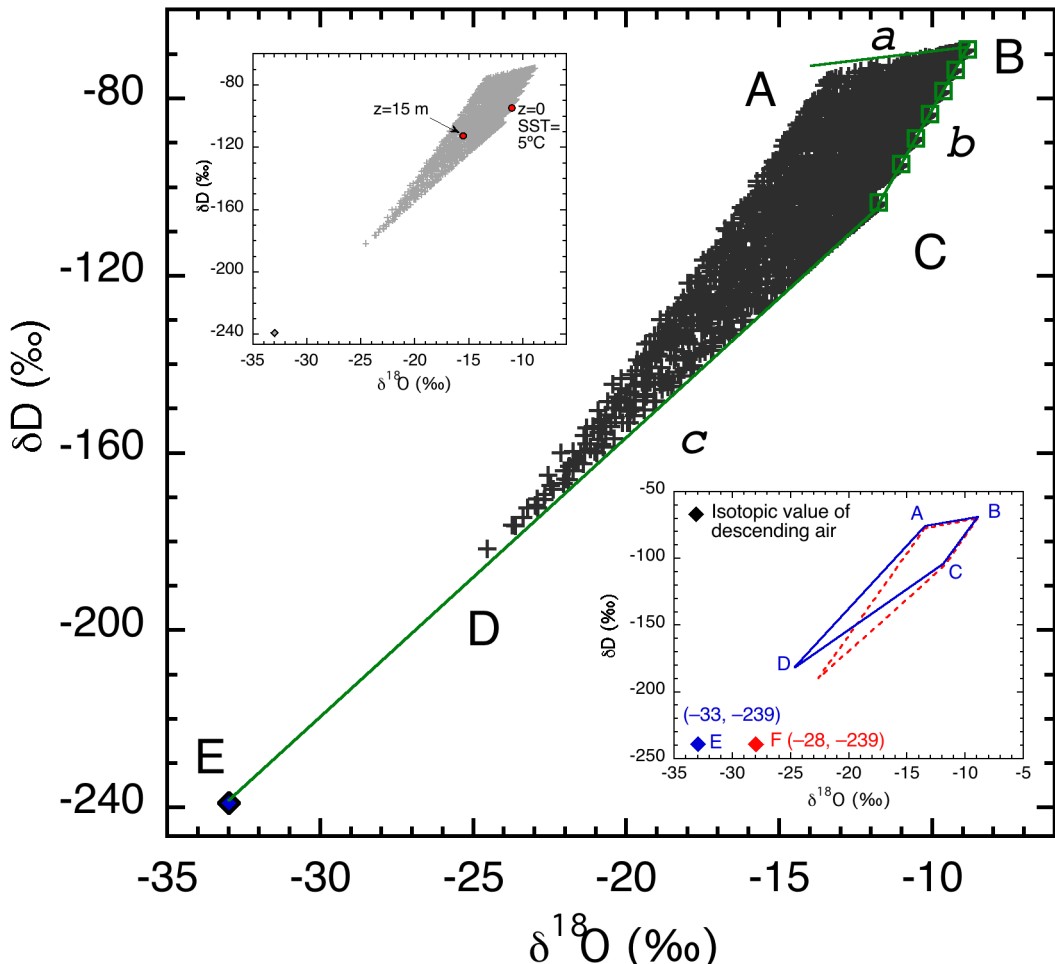

**Figure 4:** δD vs. δ¹⁸O relationship at 15 m height for 2835 model calculations (+). The output defines a quadrilateral with corners labeled by A-D. Also shown are δD and δ¹⁸O values of the descending air (E, ◆), and isotope values of vapor in equilibrium with the seawater (□, along line *b*) at SSTs of –2, 5, 10, 15, 20, 25, and 30°C from C to B, respectively. Solid lines labeled *a*, *b*, and *c* bound the theoretical limits of vapor isotopic values, with *b* being vapor in equilibrium with seawater, *c* being a mixture of vapor at the sea surface and vapor from aloft, and *a* indicating vapor produced by maximum kinetic fractionation. The upper small inset replicates in gray the 2835 points from all calculations (+), plus red solid dots to indicate vapor isotope values at 0 and 15 m above the sea surface for the run illustrated in Figure 3. The lower small inset compares two quadrilateral regions produced by assuming different δ¹⁸O values of the descending vapor. The blue area with labels A-E is identical to the main graph, and red quadrilateral corresponds to descending air composition of –28‰.

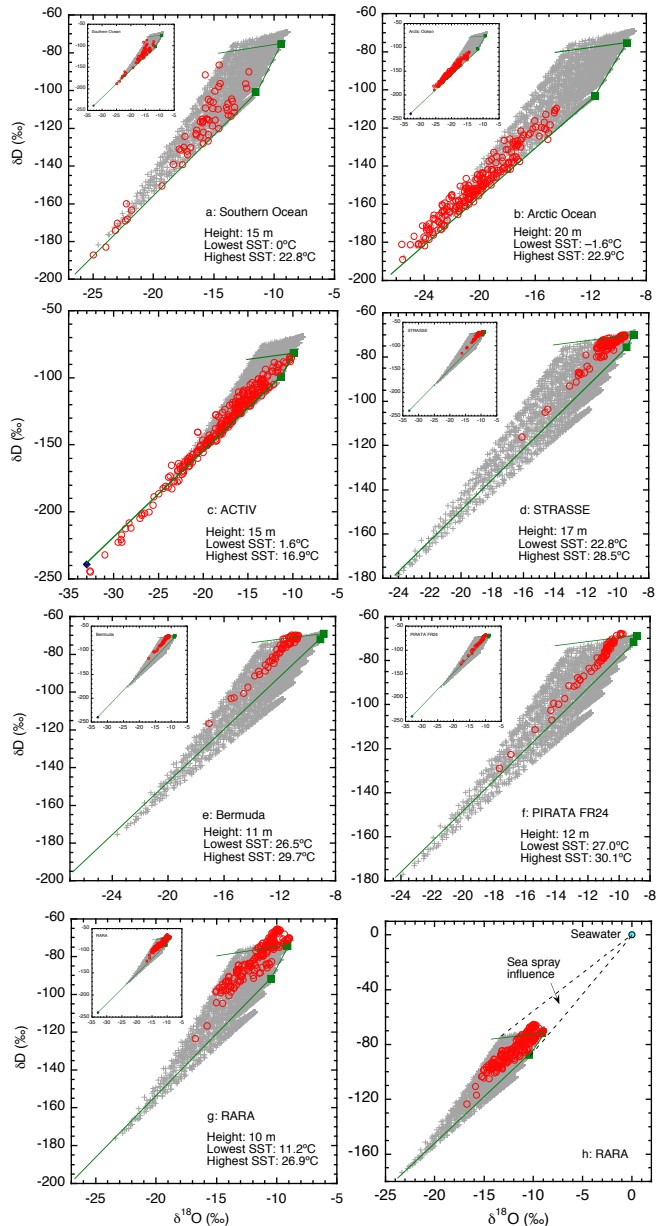

**Figure 5: Comparisons between simulated and observed isotopic ratios for each of the seven cruises (a-g), and a redraw of the RARA data with ocean water composition indicated (h). For each cruise, the simulated values are calculated at the height of the observations, indicated in the plot. If not otherwise indicated, calculated isotopic values are shown as crosses (+), the values of the descending air as a diamond (◆), and observations as red circles (O). For each cruise, vapor in equilibrium with lowest and highest SSTs is shown as two green squares (■), with the temperature values indicated in the plot. Solid lines border the theoretical limits of isotopic distributions under the cruise conditions and model assumptions. For a-g, unless the depleted, descending air is indicated in the main graph by a blue diamond (all but c) it is shown in the insert.**

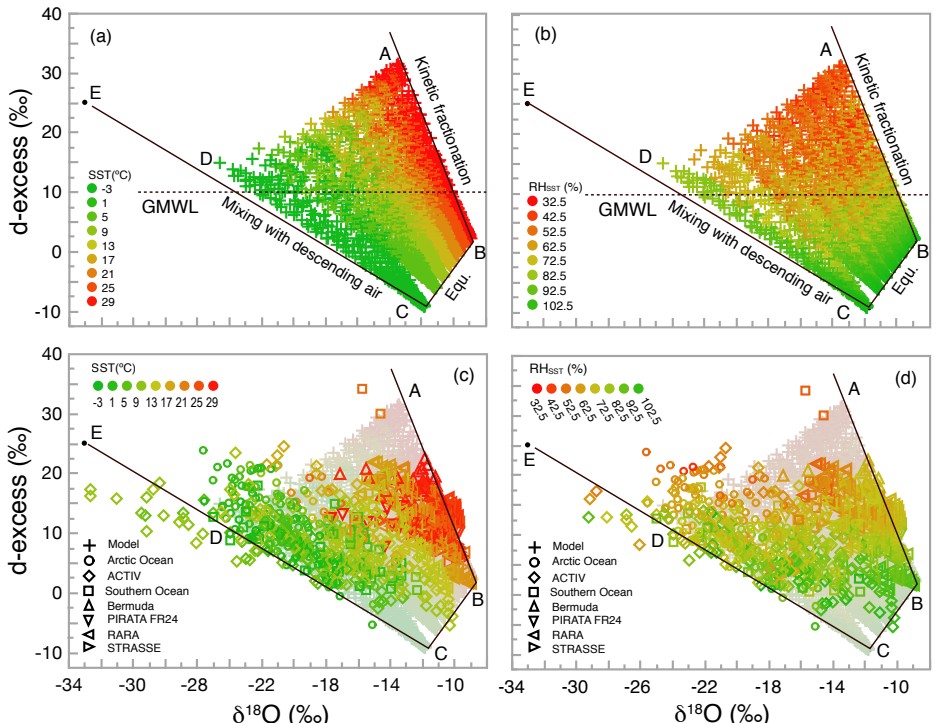

**Figure 6: Plots of d-excess vs. $\delta^{18}O$, showing relationship between d-excess and SST for the simulation (a) and model-data comparison (c), and between d-excess and $RH_{SST}$ for simulation (b) and model-data comparison (d). As in Figure 4, isotopic values are calculated at 15 m height. Points corresponding to those from A to E in Figure 4 are also shown, with point E being the isotopic composition of the descending air. Straight lines are theoretical limits for processes labeled in (a) and (b) (also discussed in section 6.1.2). The horizontal dashed lines in (a) and (b) mark the d-excess value (10) of the global meteoric water line (GMWL).**