# Peer review of "Rethinking Craig and Gordon's approach to modeling isotopic compositions of marine boundary layer vapor"

_Atmospheric Chemistry and Physics, 2018_

## Author Comment (AC1) · 25 Oct 2018

Comment from authors:
There is some error in Eqn. 16. The correct form should be

$$C(z) = \frac{C_0 ln[h_1 K_{max}] + (C_1 - C_0) ln[h_1 K_m + (K_{max} - K_m)z] - C_1 ln[h_1 K_m]}{ln\,[K_{max}/K_m]}$$

We apologize for the mistake! The code that generated the results is correct, and so all data plotted are correct.

---

## Referee Comment (RC1) · Anonymous Referee #1 · 29 Oct 2018

A one-dimensional dimensional is developed to study the variations of the water vapour isotopologues under marine boundary layer conditions. The model is used to support the analysis and interpretation of seven observational data sets. They are collected under different sea surface temperature (SST), but they are also influence by the arrival and mixing of upper atmospheric air characterized by different isotopic composition. The later processes was not explicitly modelled on the Craig-Gordon model. Model and observations are discussed with in the deltaD-delta180 space parameter. As a result, the authors identified a region that depends on the SST, the mixing of upper air and the kinetic fractionation improving the interpretation of the observations I found the research very interesting, and in particular the study on the relevance of non-local effects such as the convergence of descending air in influence the isotopic composition

near the sea surface. In addition, the model enables to cluster the seven-data sets and consequently provides a more unified explanation of the behaviour of water vapour on marine boundary layers. The paper is very well written and in my opinion deserves to be published after some of my comments are clarified.

1.- Although the Isotopic Marine Boundary Layer Model (IMBLM) has a sound physical reasoning, I believe their application and analysis can gain in depth when it is coupled to other relevant state marine boundary layer variables such as potential temperature, specific humidity and wind. In doing this coupling, they could gain independence on the determination of the eddy-diffusivity turbulent coefficient and the impact of mixing on SST. I believe the reader will appreciate a comment on the necessity of developing a model in the future to the water vapour isotopologues that it is fully coupled to the meteorological state variables.

2.- The use of an exchange coefficient for both diffusion and turbulence (Eq. 4 at page 7) is not common in meteorological models, but it I guess it is needed in studies related to the isotopologues. Km, for molecular diffusion, is normally few order of magnitude small than the turbulent diffusion, and therefore it is normally neglected. Could the authors provide a better justification on the use of Eq. 4? How do the scale the results? What is the vertical resolution needed near the sea surface?

3.- Closely connected to my previous point, normally above the see the exchange coefficient is parameterized using a roughness length that depends on the friction velocity (Charnock, 1955). This formulation can be useful to include the effects of waves and turbulence on the sea surface through a dependence on the friction velocity (page 6, lines 18-22)

4.- Why is a negative sign between the D and the advective vertical velocity at the continuity equation 6? What is the sign convection?

5.- Marine boundary layers are frequently characterized by the presence of clouds. Some of the campaigns, for instance RARA, STRASSE and PIRATE were located in

regions dominated by stratocumulus. In that respect, I would expect a discontinuity between the marine boundary layer and the free atmosphere due to the temperature inversion at cloud top. I cannot find this discontinuity in the profiles shown in Figure 3. Can they provide a more elaborated explanation on the different marine boundary layer under study?

6.- Connected to the last point, how and where is CE (Eq. 8, page 9) depicted in Figure 3

7.- Section 6.1. Could the authors justify the selection of the "few major features"?

8.- Related to the parameters selected in Table 4:

- Section 5.5. Some marine boundary layers can have larger values of the upward velocity driven by long wave radiative cooling at cloud top or the venting driven by the presence of shallow convection.

- How relevant is the height of the MCL as a variable controlling the mixing, entrainment and dilution of the isotopologues?

9.- For the sake of completeness, I believe it is convenient to include how the kinetic fractionation process (page 16, lines 15-20) is represented at the IMBL model.

10.- Page 17 (lines 18-20). Do the conditions with larger Km and lower mixing ratio of subsiding air imply well-mixed conditions? Please explain.

11- In a model in which the processes between the sea surface and MBL dynamics are fully coupled, the mixing of air with lower/higher isotopic content can have an influence on the kinetic fractionation. In other words, could the line AB in figure 4 change when the sea-MBL coupling is important?

12.- To reinforce the originality of the study, I believe the reader will appreciate a more elaborate discussion on the difference of current results with the Craig-Gordon model. For instance, what will be the results of the Craig-Gordon model in Figure 4?

СЗ

References

Charnock, H. (1955). Wind stress on a water surface. Quart. J. Roy. Meteorol. Soc., 81, 639-640.

---

## Referee Comment (RC2) · Anonymous Referee #2 · 9 Dec 2018

In the manuscript by Feng et al. a model is presented to explain the marine boundary layer water vapor isotopic composition. The authors present this as a step forward from the initial model developed more than half a century ago by Craig and Gordon.

This manuscript fails to demonstrate the usefulness of their new model to understand processes in the marine boundary layer. The authors show that they can explain the marine boundary layer water vapor isotope observations using different configurations of the parameters in their model, but they do not reflect upon what this means for our understanding of the atmospheric physical processes. To warrant publication the authors should make it clear to the reader what their model can be used for. Simply using some ad-hoc parameters to show that the model simulate observations does not expand our knowledge and understanding of the world that we live in. Specifically, the
authors should make it clear which research questions they are going to answer in this manuscript.

The authors fail to discuss developments in the use of water isotopes to understand marine boundary layer processes over the last decade. It would be an important step for potential publication of the manuscript that the authors discuss in the introduction how their new work relates itself to recent work and not just work by Craig and Gordon 1964 and Merlivat and Jouzel 1979.

The model is presented here as relieved of the need for empirically chosen values of the kinetic fractionation factor such as the k-factor in the work of Merlivat and Jouzel 1979. Instead the model introduces assumption of linear increase of diffusion from the surface to a specific height above the layer (the authors refer to this as the thickness of the von Karman layer). It is unclear in the manuscript what theoretical background or empirical observations the authors have for choosing the value of the turbulent diffusion coefficient at the interphase of the von Karman layer and convergence layer, and what foundation the authors have for deciding that the diffusion is linear in the von Karman layer. It seems that the authors replace one ad hoc parameterization with another ad hoc parameterization.

As such the manuscript could potentially be publishable, but the authors should present the manuscript for what it is: Another model of water vapor isotopologues in the marine boundary layer and not as the title suggest something which goes beyond Craig-Gordon. The text should also represent this more realistic goal of being one model among many others. Finally, the manuscript should clearly outline, why this model is useful. This could be achieved by formulating clearly outlined research questions, which the model is used to answer.

---

## Author Response (AR1)

Response to the comments of Reviewer 1.

We thank this reviewer for her/his very constructive and thoughtful comments. Many of the comments are quite insightful. Some have been addressed in this paper; others are deferred to future papers that are natural extensions of this work. In the following, we respond to each of the comments by this reviewer. For each comment, we first quote the original comment, followed by our response (indicated by ">>"), and then by citing corresponding revisions (if any) in the new manuscript (indicated by ">>>").

1. "Although the Isotopic Marine Boundary Layer Model (IMBLM) has a sound physical reasoning, I believe their application and analysis can gain in depth when it is coupled to other relevant state marine boundary layer variables such as potential temperature, specific humidity and wind. In doing this coupling, they could gain independence on the determination of the eddy-diffusivity turbulent coefficient and the impact of mixing on SST. I believe the reader will appreciate a comment on the necessity of developing a model in the future to the water vapour isotopologues that it is fully coupled to the meteorological state variables."

>> The reviewer opens an aspect of the important question of closure. As s/he points out, we closed one part of our problem by specifying the eddy-diffusivity turbulent coefficient (K, here). This limits the current model by requiring the user to look elsewhere first -- winds, stability, surface roughness, and the dependence of K on these meteorological variables – for a basis to specify K, and then use K in our model. The reviewer is certainly correct to expect that a future application that lays out a method for such coupling will be very useful to readers. Unfortunately, this coupling has been studied in too much breadth, depth and variety in boundary layer (BL) dynamics research to do it justice in this paper. Nonetheless, we concur with the reviewer in hoping that it will be combined with our model in future work. We pointed out one path to do so by citing Sheppard (1958) and Sverdrup (1946, 1951) as work that bridges use of the friction velocity  $u^*$  and von Kármán's constant  $\kappa$ , and of the coefficients  $K_m$  and b.

>>> There is no corresponding revision.

2. "The use of an exchange coefficient for both diffusion and turbulence (Eq. 4 at page 7) is not common in meteorological models, but it I guess it is needed in studies related to the isotopologues. Km, for molecular diffusion, is normally few order of magnitude small than the turbulent diffusion, and therefore it is normally neglected. Could the authors provide a better justification on the use of Eq. 4? How do the scale the results? What is the vertical resolution needed near the sea surface?"

>> The reviewer is correct; meteorological models usually include effects of molecular diffusion in the thin laminar boundary layer (LBL) implicitly in the boundary condition at the air-sea interface, and of turbulence in the atmosphere above the LBL. This is quite

reasonable for most meteorological model applications, but as the reviewer surmises, it is important for our isotopic model that the laminar layer and the turbulent part of the lower marine boundary layer (MBL) be treated consistently. We therefore use the linear form equivalent to that applied to boundary layer mixing above the LBL by Montgomery (1940], and within the LBL by Sverdrup (1946; 1951).

 $K_m$ , as the reviewer correctly notes, is orders of magnitude less than K for z>> z\*; in principle we could introduce a discontinuity at some height greater than z\*, and neglect  $K_m$  above that height. But by retaining  $K_m$  we have the continuity to solve a single differential equation governing all heights in the range [0, h1]. There is no need to make any adjustment for including the negligible  $K_m$  in the upper part of [0, h1]. (It can also be argued that including  $K_m$ , as we do, is more realistic, but admittedly the difference is negligible.)

At the same time, the model describes the variations of isotopes within the LBL [0,  $z^*$ ], in the transition region [ $z^*$ , several times  $z^*$ ] where transport changes from diffusional to turbulent, and above as one *continuous* function. Since the governing equation and its solution are continuous in [0, h1], there is essentially infinite resolution near the sea surface, and the user is free to interrogate the solution at any set of heights. To give a sense of scale, the  $z^*$  of our calculations ranges from  $10^{-3}$  to 52 cm with a median of 2.8 cm.

Also notable with our formulation is the freedom from any requirement to specify the relationship between the meteorology and kinetic fractionation below z\*, which is part of the solution. Therefore, this treatment is essentially win-win, if one is not daunted by the differential equation. We are grateful that this reviewer's question has allowed us to expand on the advantages of our approach.

>>> To give a sense of scale to those readers who have the same question as this reviewer, we added a brief discussion of the distribution of z\* to Section 6.1.2.

3. "Closely connected to my previous point, normally above the see the exchange coefficient is parameterized using a roughness length that depends on the friction velocity (Charnock, 1955). This formulation can be useful to include the effects of waves and turbulence on the sea surface through a dependence on the friction velocity (page 6, lines 18-22)."

>> We recognize that the sea-air exchange rate may be conducive to parameterization in terms of various turbulence parameters such as roughness length, friction velocity, Reynolds number, Richardson number, and many others. This is especially true in models unlike ours in which the LBL is reduced to an implicit part of the boundary conditions at the sea-air interface. In our model, Kmax plays a role similar to these other turbulence parameters by determining z\*, which in turn determines the near-surface vapor gradients and the fluxes (i.e., the sea-air exchanges). The model thus has the advantage of independence from any prior assumption about the relationships between conventional turbulence parameters and fluxes. On the other hand, our current model does not include the effects of sea surface roughness, spray (with some discussion in this manuscript) or other surface effects, so reconsideration in future papers of our LBL and the boundary conditions at z=0 might lead to improvements.

>>> No corresponding revision at this time.

4. "Why is a negative sign between the D and the advective vertical velocity at the continuity equation 6? What is the sign convection?"

>> Our sign convention is that z increases upwards. The negative sign is consistent with w (positive for upward velocity) increasing upward (positive  $\partial w/\partial z$ ) when D (convergence) is positive. In other words, the inward convergence D equals the upward velocity gradient  $\partial w/\partial z$ , for mass balance. (We're sorry if the notation D made this reviewer think of DIVergence; the letter C was already taken).

>>> We have added more explanation about the signs of D and w under Equation 6.

5. "Marine boundary layers are frequently characterized by the presence of clouds. Some of the campaigns, for instance RARA, STRASSE and PIRATE were located in regions dominated by stratocumulus. In that respect, I would expect a discontinuity between the marine boundary layer and the free atmosphere due to the temperature inversion at cloud top. I cannot find this discontinuity in the profiles shown in Figure 3. Can they provide a more elaborated explanation on the different marine boundary layer under study?"

>> The reviewer is correct that MBLs in some regions and seasons often have clouds within them. Our model is not intended to be valid in such circumstances, although their influence would likely be small at the typical measurement heights (10-20 m) on clear days. Strictly, our results should be compared only with observations with the lowest cloud base above h3, which was 1000 m in the results presented, including Figure 3. Stratocumulus (if any) above h3 would not be relevant to the model, and the inversion that often accompanies stratocumulus would be above the profiles within the model. Clouds also bring about an issue of liquid-vapor isotopic exchange, which, as we stated in the paper, is also not included in the current version of the model. By raising this question, the reviewer indicates a direction for potentially useful future work.

>>> We added this low cloud limitation to the second paragraph of Section 6.2.

6. "Connected to the last point, how and where is CE (Eq. 8, page 9) depicted in Figure 3"

>> Does the reviewer mean Figure 1 or Figure 3? In Figure 1, it is indicated along the left side, at middle height, the bold arrow headed downward (subsiding air) turns to head into the section of the MBL under study, thus becoming converging air originating externally to the study volume. The term  $C_E$  (Eq. 8, page 9) is an isotopologue

concentration ratio of this Converging External air. In Figure 3, the values of CE are not depicted. CE for H216O, HD16O and H218O can be obtained from  $r_E$  (mixing ratio),  $\delta$ D and  $\delta^{18}$ O of subsiding air". These values are given in Table 4 (–239‰ and –33‰, for  $\delta$ D and  $\delta^{18}$ O, respectively). We did not mark the isotopic values of the subsiding air in Figure 3, because they are off-scale. To expand the scale to include them would require reduction of the resolution of the near surface gradient, which is the main point of this graph. However, in response to the reviewer's comment, we have added the values of  $\delta$ D,  $\delta^{18}$ O and d-excess of the subsiding air into the caption of Figure 3. In Figure 4, the isotopic composition of the external air is located as the point E.

>>> As mentioned above, we have added the values of  $\delta D$ ,  $\delta^{18}O$  and d-excess of the subsiding air into the caption of Figure 3. This improves the readability of the figure. We thank this reviewer for pointing this out.

7. "Section 6.1. Could the authors justify the selection of the "few major features"?"

>> Each "major feature" is the focus of one of the three subsections of Section 6.1. The vertical profile (6.1.1) is to emphasize that this model is a true 1-D model, and thus different from all Craig-Gordon type models. The section is also important to emphasize the points that a) there is a strong gradient near the air-sea interface, and b) all isotopic vapor measurements by cruises are done at a selected height, which is one point taken from this profile. The other two "major features" were chosen because, in our judgement, they are the most effective for verification and for a better understanding of MBL interactions. In addition, both  $\delta D vs. \delta^{18}O$  and d-excess  $vs. \delta^{18}O$  relationships are of major importance in climate interpretations of both modern and ancient water vapor or precipitation (*e.g.*, ice cores, tree rings, etc.) isotopic variations.

>>> We added this justification to the first paragraph of Section 6.

- 8. "Related to the parameters selected in Table 4:
- Section 5.5. Some marine boundary layers can have larger values of the upward velocity driven by long wave radiative cooling at cloud top or the venting driven by the presence of shallow convection.
- How relevant is the height of the MCL as a variable controlling the mixing, entrainment and dilution of the isotopologues?"

>> - Section 5.5. We agree that larger values of *w* can occur, and that larger values can be driven by cloud top cooling or *deep* convection. However, the sensitivity of BL isotopes to *w* decreases with larger *w*. Larger *w* values would slightly increase the influence of the subsiding air properties on the isotopic ratios of the boundary layer. However, it would not change the quadrilateral boundaries (lines a, b, and c) in Figure 4, as discussed in the paper. Since this is the first IMBL model to include *w*, we chose to use moderate values. >>> To further justify the range of *w* values we used, we added two sentences to section 5.5.

>> We assume that a typo replaced "MBL" by "MCL" in the reviewer's question. The answer is that results show very little sensitivity to this height ( $h_3$ ). A full discussion of sensitivities is postponed to a follow up paper, but we would be glad to send preliminary results in response to any requests.

>>> The insensitivity to  $h_3$  has already been stated in Section 5.2, and does not need additional explanation.

9. "For the sake of completeness, I believe it is convenient to include how the kinetic fractionation process (page 16, lines 15-20) is represented at the IMBL model."

>> The kinetic process occurs solely because the molecular diffusion coefficient Km is different for each isotopologue. That's all there is to it. The amount of the kinetic effect, indicated by the deuterium excess, depends on the differences among the Km values and the transport processes in the MBL.

Craig and Gordon (1965) and its more recent successor models all require the kinetic fractionation to be an assumed function of the variables related to turbulent transport, so the kinetic fractionation factor had an *input* role. In contrast, the kinetic fractionation is an *output* diagnostic in our IMBL model.

>>> To describe the mechanism of kinetic fractionation more clearly, we added additional explanation after Eqns. 4 and 5.

10. "Page 17 (lines 18-20). Do the conditions with larger  $K_m$  and lower mixing ratio of subsiding air imply well-mixed conditions? Please explain."

>> (We assume that the reviewer intended to ask about larger  $K_{max}$  and lower  $\beta$ .) Yes, the larger  $K_{max}$  implies well-mixed conditions. Lower  $\beta$  is not directly related to well-mixed conditions. Low  $\beta$  implies very little dry air contributed from the upper atmosphere to the BL, such that the upper atmosphere isotopic composition has little leverage on the BL isotopic ratios.

>>> We added to the manuscript at the end of that paragraph (the first paragraph of Section 6.1.2) "Large  $K_{max}$  also creates well-mixed MBL, which is consistent with the simulated low isotopic gradients between the sea surface and 15 m."

11 "In a model in which the processes between the sea surface and MBL dynamics are fully coupled, the mixing of air with lower/higher isotopic content can have an influence on the kinetic fractionation. In other words, could the line AB in figure 4 change when the sea-MBL coupling is important?"

>> (Note that there is a difference between line AB and line a. The former depends on distribution of MBL conditions, but the latter is the theoretical limit.) This is an interesting question; results could change if two-way interactive coupling between the MBL and the surface layer of the ocean were included in the model. Such coupling may have at least two consequences that affect the isotopic distribution. One outcome may be a change of sea water isotopic composition (due to, e.g., evaporative enrichment), which affects equilibrium as well as kinetic fractionation in the atmosphere directly above the surface (by changing the vertical gradient of isotopic composition). The second result may be a change in SST, whose direct effect on isotopologues is to change the equilibrium fractionation. A secondary effect may be to change the vapor gradient, relative to saturation vapor composition, and the diffusion coefficient (which is temperature dependent, although the molecular diffusivity ratio is assumed to be independent of temperature).

The timescale of changes would need to be known to determine if it is necessary to consider this interaction explicitly. For a long timescale (e.g., seasonal), this IMBL could be used without modification if the user could specify changes in the sea surface boundary condition over one season, but if the conditions are known only at the beginning of the time period, the model would have to be extended to include changes occurring within the surface layer of the ocean.

>>> In order to maintain the focus of the paper, we did not add the above reply to the manuscript.

12. "To reinforce the originality of the study, I believe the reader will appreciate a more elaborate discussion on the difference of current results with the Craig-Gordon model. For instance, what will be the results of the Craig-Gordon model in Figure 4?"

>> This is a good thought. Unfortunately, there is no simple one-to-one comparison between the two models, because they have different MBL structure as well as different inputs and outputs. For example, unlike our model, the Craig-Gordon model does not yield the isotopic ratio of vapor within the boundary layer. Instead, it requires the isotopic ratio of the free atmosphere as input, and calculates only the isotopic ratio of the vapor *flux* from the sea surface. In contrast, our model calculates isotopic ratios of vapor within, and flux through, the MBL. Figure 4 shows the isotopic ratios of vapor in the MBL, but the Craig-Gordon model does not provide the output required for an equivalent plot.

The two models can still be compared if some assumptions are made. One way to accomplish this is by using the so called "closure assumption" for the Craig and Gordon type model proposed by Merlivat and Jouzel (1979), where the vapor isotopic ratio in the MBL is assumed to be equal to the isotopic ratio of the vapor flux. We eschew this closure assumption since it was demonstrated to be numerically incorrect 20 years ago (Jouzel and Koster, 1996). So any discrepancy in a comparison between the two sets of output would not provide much insight.

Another way to compare the two models is to use a particular subset of observed isotopic ratios above the sea as the input parameter to the Craig-Gordon model and calculate the isotopic ratio of the vapor flux (remember Craig-Gordon model calculates only this flux). This flux can then be compared to the flux calculated by our model, provided that the isotopic ratio profile generated by our model agrees with the observed value at the observed height. We have already made such a comparison in a 2018 Goldschmidt conference presentation (Welp et al., 2018). The conditions for this experiment were close to saturation, where Craig and Gordon type models are least reliable, so poor agreement was obtained as expected. This could be interesting to extend in future work, but is difficult to include in this manuscript for two reasons. First, there are few direct observations of isotopic ratio of vapor flux at the sea surface. Therefore, there is no target (right answer, or empirical observations) to compare with. A better way of doing a comparison would be to conduct vapor observations near an eddy correlation tower equipped with isotopic measurements (still a challenge for today's technology), or to make vapor measurements from multiple heights. This way, there would be some guidance about what the true flux might be. We are in the process of making an effort in this direction using large lakes; the results will be reported elsewhere. Second, in addition to the difficulty pointed out above, we would have to take a significant diversion from the main point of the paper in order to include such a comparison. This is because the comparative calculations have to be informed by much more detailed MBL conditions associated with a cruise or a section of a cruise, including vertical velocity,  $K_{max}$ , etc. The current paper is already too long, and a detailed application of the model is better treated in separate contributions.

>>> No corresponding revision at this time.

**References Cited:**

Craig, H., and Gordon, L. I.: Deuterium and oxygen-18 variations in the ocean and marine atmosphere, in: Stable Isotopes in Oceanographic Studies and Paleotemperatures, edited by: Tongiorgi, E., Spoleto, Italy, 9-130, 1965.

Jouzel, J., and Koster, R. D.: A reconsideration of the initial conditions used for stable water isotope models, Journal of Geophysical Research: Atmospheres, 101, 22933-22938, 10.1029/96jd02362, 1996.

Merlivat, L., and Jouzel, J.: Global climatic interpretation of the deuterium-oxygen 18 relationship for precipitation, Journal of Geophysical Research, 84, 5029-5033, 1979.

Montgomery, R. B.: Observations of vertical humidity distribution above the ocean surface and their relation to evaporation, Papers in Physical Oceanography and Meteorology (MIT and WHOI), 7, 1940.

Sheppard, P. A.: Transfer across the earth's surface and through the air above, Q. J. R. Meteorol. Soc., 84, 205-224, 1958.

Sverdrup, H. U.: The humidity gradient over the sea surface, Journal of Meteorology, 3, 1-8, 1946.

Sverdrup, H. U.: Evaporation from the oceans, in: Compendium of Meteorology, edited by: Malone, T. F., American Meteorological Society, 1071–1081, 1951.

Welp, L., Meyer, A., Griffis, T., Feng, X., and Posmentier, E. S.: In-situ observations of water vapor isotopes in near surface air over Lakes Superior and Michigan, Goldschmidt Conference, Boston MA, August 12-17, 2018.

Response to the comments of Reviewer 2.

This reviewer has a few comments about the relevance of this paper. In several cases, we believe that the answer was already in the manuscript. Here, we try to explain some of the context of this model slightly differently in hopes that it will make some of the points of the paper clearer.

Like our response to Reviewer 1, we first quote the original comment, followed by our response (using the prompt ">>"), and then citing corresponding revisions (if any) in the new manuscript (using the prompt ">>>").

1. "This manuscript fails to demonstrate the usefulness of their new model to understand processes in the marine boundary layer. The authors show that they can explain the marine boundary layer water vapor isotope observations using different configurations of the parameters in their model, but they do not reflect upon what this means for our understanding of the atmospheric physical processes. To warrant publication the authors should make it clear to the reader what their model can be used for. Simply using some ad-hoc parameters to show that the model simulate observations does not expand our knowledge and understanding of the world that we live in. Specifically, the authors should make it clear which research questions they are going to answer in this manuscript."

>> We do not fully understand this comment, i.e., what this reviewer had in mind about "the usefulness of their new model" in understanding "processes in the marine boundary layer". We indicate where in the paper we provided context of this model, where we discussed the processes, and where we stated the "the usefulness" of the model, in some cases adding additional details and specific examples to illustrate how the model will aid in the understanding of "processes in the marine boundary layer".

Most context was given in the Introduction, where we discussed why we developed this model. Part of the reason was to overcome limitations of the Craig and Gordon model for understanding marine boundary layer (MBL) processes, for connecting the MBL with Rayleigh processes in the free atmosphere, and for simulating new observations, particularly vapor isotopic measurements. A significant part of our motivation was to create a model capable of exploring how meteorological processes and conditions in the MBL (e.g., convergence of external air) affect the isotopic compositions of air masses that produce precipitation.

Regarding the use of the model to understand boundary layer (BL) processes, one important discussion is the explanation of Figure 4, which summarizes the isotopic systematics of the boundary layer vapor produced by our model. We identified three processes that define the quadrilateral shape of the  $\delta D$  vs.  $\delta^{18}O$  distribution, namely equilibrium isotopic fractionation, kinetic isotopic fractionation, and mixing of the boundary layer vapor subsided from the upper atmosphere. The relative importance of these processes depends on MBL conditions that are represented by a set of parameters, including turbulence intensity K(z), vertical velocity w, and the properties

of the subsiding air ( $C_E$ ), each having a specific physical meaning. As stated in the manuscript, a full sensitivity test will be given in a separate contribution due to length limitation of this manuscript, but we hope that our discussion is sufficient at this point.

The specific use of our model is summarized in the last paragraph of the Conclusions (Section 7) as quoted below:

'The IMBL model can be used in a number of ways. First, numerical experiments with the model help to better understand the effects of boundary layer processes and climatic conditions on isotopic compositions of vapor within and vapor fluxes through the MBL. A second application could be to investigate how temporal and spatial differences in moisture source regions affect the isotopic composition of remote precipitation under both modern as well as paleo-climate conditions. Third, it is important to investigate the relationship between MBL isotopes and evaporation rate and, perhaps, to develop methods to measure the latter indirectly from simultaneous observations of isotopes and meteorological conditions. Finally, the understanding gained from IMBL model simulations can be used to improve the representation of MBL processes in isotope enabled GCMs.'

>>> We modified the above paragraph by adding a few specific scientific questions that can be investigated with our model. We hope this is consistent with what the reviewer expected. The new paragraph is below, with highlighted text indicating newly added text.

'The IMBL model can be used in a number of ways. First, numerical experiments with the model help to better understand the effects of boundary layer processes and climatic conditions on isotopic compositions of vapor within and vapor fluxes through the MBL. For example, one may investigate how boundary layer stability, turbulence conditions, vertical velocity, convergence, and upper atmospheric moisture affect MBL isotopic distributions and how these effects change with space and time. A second application could be to investigate how temporal and spatial meteorological differences in moisture source regions affect the isotopic composition of remote precipitation under both modern as well as paleo-climate conditions. In this application, the IMBL model can be coupled with a Rayleigh distillation model to simulate isotopic evolution of vapor and/or precipitation from moisture source to a precipitation site. These simulations can be particularly powerful if also used in conjunction with a Lagrangian back trajectory program to identify moisture source areas for a site of interest. Third, it is important to investigate the relationship between MBL isotopes and evaporation rate and, perhaps, to develop methods to measure the latter indirectly from simultaneous observations of isotopes and meteorological conditions. Since this IMBL model calculates the flux of each isotopologue (rather than just their ratios), it yields the evaporation rate. This opens up the possibility of using isotopic measurements to quantify evaporation rates under various boundary layer conditions. Finally, the understanding gained from IMBL model simulations can be used to improve the representation of MBL processes in isotope-enabled GCMs.'

2. "The authors fail to discuss developments in the use of water isotopes to understand marine boundary layer processes over the last decade. It would be an important step for potential publication of the manuscript that the authors discuss in the introduction how their new work relates itself to recent work and not just work by Craig and Gordon 1964 and Merlivat and Jouzel 1979." >>In terms of isotopic boundary layer model development, we are not aware of any significant change since Merlivat and Jouzel (1979). To our knowledge, the recent work has added significantly to the *number of vapor measurements* in the marine boundary layer, but related modeling of the data is still based on the earlier work by Craig and Gordon, (1965), and Merlivat and Jouzel (1979). In the introduction, we explained the difference between our model and the earlier models, in terms of conceptualization of the boundary layer structure and processes. If this reviewer thinks that we are unaware of models fundamentally different from what we call "Craig and Gordon type models" (other than isotope enabled GCMs) for the use of boundary layer isotopic distributions, please give us specific references.

**>>> No revision is made.**

3. "The model is presented here as relieved of the need for empirically chosen values of the kinetic fractionation factor such as the k-factor in the work of Merlivat and Jouzel 1979. Instead the model introduces assumption of linear increase of diffusion from the surface to a specific height above the layer (the authors refer to this as the thickness of the von Karman layer). It is unclear in the manuscript what theoretical background or empirical observations the authors have for choosing the value of the turbulent diffusion coefficient at the interphase of the von Karman layer and convergence layer, and what foundation the authors have for deciding that the diffusion is linear in the von Karman layer. It seems that the authors replace one ad hoc parameterization with another ad hoc parameterization."

>> This comment has several points. We break it down into three questions (although not in the same order as they are given) and address them separately below: A) What is the basis of our assumption that the eddy diffusion coefficient increases linearly with height in the lower layer of the MBL? B) By using Eq. (4) ( $K=K_m+bz$ ), do we simply "replace one ad hoc parameterization with another ad hoc parameterization"? C) how do we choose the value of  $K_{max}$ ?

>> A) The "law of the wall", first published by von Kármán (1930), states that, in the layer close to the boundary, turbulent velocity increases logarithmically with height (i.e., the logarithmic wind speed profile). Actual wind speed profiles have, of course, been observed. The logarithmic profiles occur in the lowest 5-10% of the boundary layer (our lower layer) under statically neutral conditions. Some deviations may occur under either stable or strongly convective conditions in opposite directions, so the logarithmic profile is a good representation of the median situation. If using a first order local turbulence closure scheme, or K-theory, (*i.e.*, the flux is proportional to the gradient) for the turbulent transport, which is what we used in this paper, this logarithmic wind speed profile is mathematically consistent with a linear increase of K (eddy diffusion coefficient) with height (z). K-theory has been used in all Craig-Gordon type models. Our change makes the model truly one-dimensional, with K continuously increasing with

*z* in the lower layer. By making these changes, we significantly alter the representation of kinetic isotopic fractionation caused by molecular diffusion close to the water-air interface (See B below). We did not consider it necessary to include significant background of textbook boundary layer meteorology in the paper. Interested readers can consult, e.g., Stull (1988).

**>>> No revision for A).**

>> B) This reviewer thinks that our representation of kinetic fractionation may be just a replacement of "one ad hoc parameterization with another ad hoc parameterization". We strongly disagree. Yes, we do have to parameterize or specify  $K_{max}$  to use Eq. (4), rather than specifying the kinetic fractionation factor (k in Merlivat and Jouzel, 1979). However,  $K_{max}$  is a boundary layer dynamics parameter that already exists in boundary layer dynamics literature. The study of K(z) has much greater "breadth, depth and variety in boundary layer (BL) dynamics" (quote from our response to Reviewer 1) and is based on a much richer set of observations than is the kinetic fractionation factor, k. More importantly, by making kinetic fractionation a function of K(z), our model is capable of exploring how BL meteorological conditions affect the kinetic isotopic fractionation, one important improvement over Craig and Gordon type models (and, by the way, one example of using the model to study processes). The above points also amplify our response to A), which brings up the question of what indeed the improvement of our model over earlier models is in terms of how it represents kinetic fractionation. Further, they explain more, here in a slightly different context, our response to Reviewer 1 (points 2 and 9).

In addition to the above advantage, there are other very important benefits from using Equation (4). First, our formulation allows our model to compute fluxes of isotopologues, not just their ratios. Second, it allows the model to compute the elusive value of sea surface evaporation. Third, it allows us *"to investigate the relationship between MBL isotopes and evaporation rate and, perhaps, to develop methods to measure the latter indirectly from simultaneous observations of isotopes and meteorological conditions"* (quote from the manuscript and response 1 to this reviewer). In fact, we have already used a different version of this model to study lake evaporation rates (Feng, et al. 2016). Furthermore, with *K* a continuous function of *z*, our model is truly one-dimensional, which allows vertical isotopic profiles to be predicted and compared with isotopic observations at multiple heights and with any resolution. These benefits all provide additional power of the model for studying boundary layer processes. Craig and Gordon type models, on the other hand, by parameterizing or specifying *k*, essentially sacrifice any of these possibilities. So our parameterization is not just another inconsequential alternative.

Related to our answer above, we note that our formulation of the kinetic fractionation provides much greater potential to be coupled with various types of boundary layer dynamics models (including GCMs) than previous models. In most of these dynamics models, the representation of turbulence transport is already established. The transport coefficient may or may not be a linear function of *z*, but

adding isotopologues to any of these models would be very easy to do using  $K(z)=K_m+K_T(z)$ , where  $K_T(z)$  is the turbulent diffusion coefficient as a function of z. Historically, BL isotopic models and their output have been of interest mainly to the isotope hydrology and paleoclimate communities. We think, however, that isotopic measurements can be a powerful tool to investigate general boundary layer processes, contributing to advancement of boundary layer meteorology. This model provides one bridge towards that goal.

>>> We thank both reviewers for questioning our kinetic fractionation formulation, which gives us an opportunity to explain its benefits more fully in this open discussion setting. A brief summary of the above explanation is incorporated into the manuscript as the last paragraph of Section 2.1.

>> C) How did we choose  $K_{max}$ ? We assume that this is the question meant by the reviewer in the original comment "It is unclear in the manuscript what theoretical background or empirical observations the authors have for choosing the value of the turbulent diffusion coefficient at the interphase of the von Karman layer and convergence layer...". The information is already in the manuscript. We explained the K(z) profile, in the third paragraph under Section 2, and illustrated it in Figure 1. We cited O'Brien (1970) for a typical K profile in the BL, which is also cited by Stull (1988). We present Stull's graph below for additional information to this reviewer and to readers with a similar question, but do not think it necessary to include it in the manuscript.

From: Stull, R. B.: An introduction to boundary layer meteorology, Springer Science & Business Media, 1988, page 210.

Fig. 6.2 Typical variation of eddy viscosity, K, with height in the boundary layer. After O'Brien (1970).

The choice of  $K_{max}$  is discussed in Section 5.3, with citations to the literature. We do not consider it necessary to repeat the discussion here. We note, however, that we do not use only one value, but explore the effect of varying  $K_{max}$  over a wide range on the isotopic compositions of the BL. This is only one example of how the model has

already improved our understanding of MBL processes, an understanding that Reviewer 2 sought but did not find in the paper (See 1, above). If the reviewer has more specific questions, we are happy to answer.

>>> No revisions for C).

4. As such the manuscript could potentially be publishable, but the authors should present the manuscript for what it is: Another model of water vapor isotopologues in the marine boundary layer and not as the title suggest something which goes beyond Craig- Gordon. The text should also represent this more realistic goal of being one model among many others. Finally, the manuscript should clearly outline, why this model is useful. This could be achieved by formulating clearly outlined research questions, which the model is used to answer.

>> We hope that we have already addressed most concerns of this reviewer above. In terms of the title of this paper, this model extends the Craig-Gordon type models in several new directions. It adds convergence and vertical advection to the model; it does not use an assumed parameterized kinetic fractionation; it is continuous over height and thus resolves scales from less than  $z^*$  to  $h_3$ , a range of a factor of over  $10^5$ . So the suggestion in the title is quite well justified, and this should have been quite obvious in the original manuscript. Very detailed explanations about how this model differs from Craig-Gordon type models, and what motivated us to incorporate the new features are given in Introduction of the manuscript. If this reviewer thinks that there are many other (non-Craig-Gordon type) models that do what our IMBL model does, please provide us specific references.

>>> No revision is made.

**References Cited:**

[revised manuscript text omitted]

---

## Author Response (AR2)

Dear Dr. Rockman,

With this cover letter, we are submitting the revised manuscript (acp-2018-709), retitled as
 **"Rethinking Craig and Gordon's approach to modeling isotopic compositions of marine boundary layer vapor"**

Thank you so much for your support of our work! We appreciate both your understanding of our arguments and your balanced judgment of our work. We also understand your concern that we may not have sufficiently and fairly acknowledged the work conducted before ours, particularly but not exclusively the work of Benetti and colleagues.

Before we address your suggestion of comparing our work with Benetti et al.'s work, we would like to first explain why in general we find it difficult to do site-specific comparisons in this paper. As you know, in several reviews (including the earlier reviews when the paper was submitted to JGR), we were asked to do a detailed comparison or a simulation for a specific site. These are difficult to do, for the reason we articulated in our original response to you:

> "the purpose of this work is not to conduct detailed simulations to match specific data sets; doing so
> would require much more comprehensive consideration of site specific conditions, including values
> of SST, sea water isotopic ratios (for which we also used only one set of values), air temperature and
> RH, wind direction and speed, sea surface roughness, vertical air stability, etc. The manuscript is
> already too long, and we feel that it is better to leave these considerations to later papers that carry
> out simulations."

Considering that many readers may have the same thoughts as you and the reviewers when reading this paper, we added a paragraph to express the need for doing so, even though we postpone it to later investigations. The following paragraph is added to the end of section 6.1.

> To end this section, we point out that our model-observation comparisons are focused on identifying major processes controlling large-scale isotopic distributions of water vapor. These general comparisons should be followed by simulations specific to given sites over given observation time windows, which would require narrowing down model parameterizations according to the conditions where and when data were collected. For example, the SST, water isotopic values, vertical velocity, $K_{max}$, properties of descending air should all be obtained/estimated either from observations or from reanalysis products. Such site- and time-specific simulations will allow identifying relative importance of various processes and will lead to an understanding of how the relative contributions of each process vary over space and time. Since such work requires a particular context for each data set, we postpone it to future investigations.

Specifically regarding comparison with the work by Benetti et al., we have additional reasons why we are reluctant to do so. In an effort to respond to your request, we re-read Benetti et al's papers and spent several hours discussing their work. While Benetti et al's work is creative, it lacks rigor and there are conceptual inconsistencies in their approach that we do not want to expose openly. We give one example below to show why comparison with Benetti et al.'s work is so difficult.

Benetti *et al.* (2015) showed that the fraction of tropospheric influence is 20-60%. Can we compare that result with ours (<10%)? There are several problems.
1. The atmospheric end members are different. They used 790 hPa air, while we used ~500 hPa air.
2. The two approaches apply to very different scenarios in the marine boundary layer (for example, descending vs. ascending air).
3. Benetti et al.'s definition of $\gamma$, the fraction of troposphere vapor present at the measurement height (e.g., 17 m), is not reconciled with how $\gamma$ is used in the isotopic mixing relationship. In addition, the mixing relationship is not rigorously derived. We had a long discussion about under what conditions it would be correct, and concluded that the corresponding meteorological scenario is not realistic.

Hence, we have chosen to acknowledge Benetti et al.'s work qualitatively, but not to compare results quantitatively. Some qualitative acknowledgements were already incorporated into the last version of the manuscript. We added another one in the 5th paragraph of the Conclusions section:

> The influence of free troposphere vapor on the d-excess of the boundary layer vapor has also been demonstrated by Benetti et al. (2015, 2018) via a C-G type approach. Quantification of this influence under various meteorological scenarios should be an important objective for future investigation in order to use d-excess for ice core studies.

We have also removed the criticism that their model is still based on the closure assumption.

> Such influence of upper atmosphere air on the boundary layer has been recognized by Benetti et al., (2015, 2018), although for regions of quiescent subsidence that our model does not treat. Furthermore, their conceptual treatment is still based on the closure assumption.

Our second intense discussion this round concerned the title, which is now **"Rethinking Craig and Gordon's approach to modeling isotopic compositions of marine boundary layer vapor"**

Craig and Gordon's original paper, while long and difficult to parse, is an outstanding work. It is not only original and a classic of the 20th century, but also remarkably rigorous; they spelled out and justified every assumption they made, and the paper is logically complete. Their model has been widely applied and extended by others. Unfortunately, some of the rigor and clarity of Craig and Gordon's assumptions was lost in some of that later work. The closure assumption is just one example. In the same spirit as Craig and Gordon, we intended to present our model with the same rigor by explicitly stating each model assumption, and to explain how our model extended their conceptual structure. This was the reason behind our choice of the earlier title.

We acknowledge that "Beyond Craig and Gordon" is too broad and could lead to a perception of arrogance/oversell, however unintended. Craig and Gordon's work has been used for an incredibly wide array of applications, including the ocean boundary layer, lake evaporation, and transpiration, but our model only deals with parts of the marine boundary layer of the atmosphere. For this reason, our title should be more specific.

On the other hand, we also strongly feel that our work is really a systematic rethinking of Craig and Gordon's model structure. Benetti et al. (2015) did not change Craig and Gordon; Craig and Gordon did not say that the troposphere has no influence on the boundary layer. They only required that the isotopic composition of the free atmosphere be specified in order to calculate the flux ratio. Benetti et al. (2015) specify this value as the composition of 790 hPa air mixed with evaporative flux. On the other hand, the work by Wei et al. (2018) goes way beyond Craig and Gordon in treating a much broader system. (We have a lot of respect for this group, although we did not catch this new reference -- thank you for bringing it to our attention! This work is now cited in a couple of places in the revised manuscript.) This is an elegant, distinct presentation of a very comprehensive model that includes many dynamics in the boundary layer (including the terrestrial PBL) that we do not have in our model. Nonetheless, for evaporation, Wei et al. simply incorporated Craig and Gordon's model.

To make it clearer that our model is a rethinking and extension of the C-G model, we added the following text in the Introduction as we explained the changes we make.

> "…These three model requirements obviously require rethinking the C-G approach to constructing a model, and draw a sharp divergence between our model and the models of C-G and its extensions (e.g., Jouzel and Merlivat, 1979, and Benetti et al, 2015)." and "…The fourth and fifth major changes further sharpen the distinction between our model and the C-G type models."

We hope that you consider this title more appropriate for the scope of our work presented in this paper. Thanks again for your efforts and discerning management of our paper, reviews, and revisions. Please find the attached response to Reviewer 2 below. We also thank this reviewer for his/her efforts to help improve our manuscript!

Sincerely,
Xiahong Feng

**Response to Reviewer 2**

We recognize that this reviewer is not satisfied with our response to his criticisms. In our opinion, most of these disagreements are in the realm of normal scientific disagreements among researchers. Over time, the paper should generate intensive discussions and debate. However, we would like to respond to one comment in particular, that our paper does not answer a scientific question.

While we do not agree with the general premise that scientific contributions must always take the form of answering a question, it is nonetheless the case that we DO answer scientific questions in this paper. One of them is "how does the isotopic composition of vapor that is delivered to the free troposphere depend on boundary layer meteorological conditions?" In this case, we pooled all marine boundary layer observations we could find, and described their distribution in the $\delta D$-$\delta^{18}O$ and d-excess-$\delta^{18}O$ spaces. Using the model, we discussed how the area and points within it are controlled by various processes with different relative importances. This reviewer considers this old knowledge, although we have never seen the word "quadrilateral" used in describing isotopic distributions, nor have we read any systematic discussions about the associated mechanisms behind the distribution.

To make our contribution clearer, we added a few lines at the end of Section 6.1.1 (the text highlighted in yellow below is new). In doing this we hope to make it clear that the isotopic distribution mechanisms we discuss apply directly to understanding the isotopic distribution of vapor delivered to the free troposphere – one of the major motivations of this work.

> "Most in situ observations are conducted at a fixed height above the sea surface. The seven cruise data sets (Table 3) were collected at heights between 10 and 20 m. In these cases, each measurement represents an air sample at a given height along a vertical profile. As shown in the calculation depicted in Figure 3, isotopic gradients are greatest near the sea surface; in this example, over just 15 m (which is only 1.5% of the total height of the MBL) $\delta^{18}O$, $\delta D$ and d-excess achieve 58, 43, and 88%, respectively, of the change toward the relatively constant values between $h_2$ and $h_3$ (650-1000 m). Above 10 m, isotopic change with height is relatively slow. For example, in this particular calculation, at 15 m the $\delta^{18}O$, $\delta D$, and d-excess values are –15.6, –112.6‰, and 12.2‰, respectively; they change by only 0.50, 3.56, and 0.40‰, respectively, between 10 and 20 m. Consequently, the isotopic variations between 10 to 20 m to be discussed in the upcoming sections can be viewed as approximating the isotopic variations of vapor delivered to the free troposphere. If the actual vapor isotopic ratios of an air mass to initiate a Rayleigh process are desired, the values at $h_3$ should be used."

This reviewer is also not pleased with our title. We have acknowledged to the editor that it can be seen as claiming too much significance. A new title is now used.

[revised manuscript text omitted]